# A Cas-BCAR3 co-regulatory circuit controls lamellipodia dynamics

**Elizabeth M Steenkiste[1,2], Jason D Berndt[1†‡], Carissa Pilling[1,2†], Christopher Simpkins[1], Jonathan A Cooper[1,2]\***

[1]Division of Basic Sciences, Fred Hutchinson Cancer Research Center, Seattle, United States; [2]Molecular and Cellular Biology Program, University of Washington, Seattle, United States

**Abstract** Integrin adhesion complexes regulate cytoskeletal dynamics during cell migration. Adhesion activates phosphorylation of integrin-associated signaling proteins, including Cas (p130Cas, BCAR1), by Src-family kinases. Cas regulates leading-edge protrusion and migration in cooperation with its binding partner, BCAR3. However, it has been unclear how Cas and BCAR3 cooperate. Here, using normal epithelial cells, we find that BCAR3 localization to integrin adhesions requires Cas. In return, Cas phosphorylation, as well as lamellipodia dynamics and cell migration, requires BCAR3. These functions require the BCAR3 SH2 domain and a specific phosphorylation site, Tyr 117, that is also required for BCAR3 downregulation by the ubiquitin-proteasome system. These findings place BCAR3 in a co-regulatory positive-feedback circuit with Cas, with BCAR3 requiring Cas for localization and Cas requiring BCAR3 for activation and downstream signaling. The use of a single phosphorylation site in BCAR3 for activation and degradation ensures reliable negative feedback by the ubiquitin-proteasome system.

**\*For correspondence:**
jcooper@fhcrc.org

†These authors contributed equally to this work

Present address: ‡Seagen, Bothell, United States

## Introduction

Animal cells migrate by adhesive crawling or amoeboid blebbing (*Trepat et al., 2012*). During crawling, transmembrane receptors called integrins provide attachment to the extracellular matrix and organize the actin cytoskeleton (*Bachir et al., 2017*; *Legate et al., 2009*). Integrin engagement stimulates protrusion of a dynamic leading lamellipodium. Inside the lamellipodium, rearward flowing actin engages with integrin-associated proteins such as talin and vinculin, forming catch bonds, clustering the integrins, and recruiting additional regulatory and scaffold proteins to form transient structures called nascent adhesions (*Case and Waterman, 2015*; *del Rio et al., 2009*; *Galbraith et al., 2002*; *Partridge and Marcantonio, 2006*; *Puklin-Faucher and Sheetz, 2009*; *Tadokoro et al., 2003*). Most nascent adhesions are short-lived, but some mature into focal adhesions at the base of the lamellipodium, anchoring actin stress fibers and resisting the rearward actin flow to increase lamellipodium protrusion. While many aspects of cell migration can be explained by biomechanics, integrin adhesions also activate biochemical signaling molecules, including focal adhesion kinase (FAK), Src-family kinases (SFKs), and small GTPases (*Burridge et al., 1992*; *Huang et al., 1993*; *Miyamoto et al., 1995*; *Schaller et al., 1992*). Some FAK and SFK-dependent phosphorylations regulate adhesion assembly (*Pasapera et al., 2010*; *Stutchbury et al., 2017*; *Zaidel-Bar et al., 2007*), while others coordinate adhesion with lamellipodium dynamics and other aspects of cell biology, such as cell survival (*Mitra and Schlaepfer, 2006*).

One important adhesion-regulated signaling protein is Cas (Crk-associated substrate, also called p130Cas or BCAR1), which plays critical roles in lamellipodium protrusion, membrane ruffling, adhesion assembly, adhesion turnover, resistance to anoikis, and malignant transformation (*Honda et al., 1999*; *Sanders and Basson, 2005*; *Sharma and Mayer, 2008*; *Webb et al., 2004*; reviews: *Defilippi et al., 2006*; *Tikhmyanova et al., 2010*). Cas is activated when SFKs catalyze tyrosine

phosphorylation (pY) of multiple YxxP sites in an unstructured region near the N terminus, known as the 'substrate domain' (see *Figure 1a* for a diagram of Cas structure). Phosphorylation recruits adaptor proteins Crk and CrkL together with guanine nucleotide exchange factors (GEFs), C3G and DOCK180, that activate Rap1 and Rac1, respectively (*Fonseca et al., 2004*; *Goldberg et al., 2003*; *Hasegawa et al., 1996*; *Kiyokawa et al., 1998*; *Klemke et al., 1998*; *Sakai et al., 1994*; *Sakai et al., 1997*; *Tanaka et al., 1994*). After activation, Rap1 increases talin recruitment to integrins to promote adhesion maturation (*Boettner and Van Aelst, 2009*), while Rac1 stimulates polymerization of branched actin structures by WAVE, leading to lamellipodial protrusion and ruffling (*Cheresh et al., 1999*; *DeMali et al., 2002*; *Ridley, 2001*; *Ridley et al., 1992*; *Stradal et al., 2004*). Cas also stimulates focal adhesion turnover as the cell moves forwards (*Ren et al., 2000*; *Rottner et al., 1999*; *Webb et al., 2004*). Thus, Cas is part of a self-reinforcing cycle in which actin polymerization stimulates Cas phosphorylation, which leads to more actin polymerization, lamellipodium protrusion, membrane ruffling, and focal adhesion turnover.

While the consequences of Cas phosphorylation are well-understood, it is less clear how Cas phosphorylation is regulated. Cas phosphorylation requires integrin-dependent actin polymerization and an intact actin cytoskeleton (*Vuori et al., 1996*; *Vuori and Ruoslahti, 1995*; *Zhang et al., 2014*; *Zhao et al., 2016*). In vitro, Cas phosphorylation by SFKs is increased when the substrate domain is physically extended, suggesting that Cas may be a mechanosensor (*Hotta et al., 2014*; *Sawada et al., 2006*; *Tamada et al., 2004*). Cas contains N- and C-terminal domains that associate with integrin adhesions and could be involved in extending the substrate domain (*Braniš et al., 2017*; *Donato et al., 2010*; *Harte et al., 1996*; *Nakamoto et al., 1997*; *Nojima et al., 1995*; *Vuori and Ruoslahti, 1995*; *Figure 1a*). At the N terminus, an SH3 domain binds adhesion proteins vinculin and FAK (*Harte et al., 1996*; *Janoštiak et al., 2014*; *Polte and Hanks, 1995*), while at the C-terminus, a CCH domain binds to adhesion proteins paxillin and ajuba (*Pratt et al., 2005*; *Yi et al., 2002*). In addition, the Cas SH3 domain binds to N-WASP via FAK, and N-WASP stimulates actin polymerization and Cas phosphorylation (*Kostic and Sheetz, 2006*; *Wu et al., 2004*; *Zhang et al., 2014*). Thus, Cas may be activated by actin flow or when the substrate domain is extended by forces acting on the Cas N- and C-terminal domains.

Cas is also subject to negative feedback. Phosphorylation of a YDYV sequence near the CCH domain mediates pY-Cas degradation (*Teckchandani et al., 2014*). This phosphosite binds to the SH2 domain of SOCS6 (suppressor of cytokine signaling 6), recruiting CRL5 (Cullin 5-RING-ligase) and targeting Cas to the ubiquitin-proteasome system. SOCS6 co-localizes with pY-Cas in adhesions at the leading edge of migrating cells where it inhibits adhesion disassembly (*Teckchandani and Cooper, 2016*). Mutation of the YDYV sequence or knockdown of SOCS6 or the CRL5 scaffold, Cullin 5 (Cul5), stabilizes Cas and increases adhesion disassembly, lamellipodia protrusion and ruffling (*Teckchandani and Cooper, 2016*; *Teckchandani et al., 2014*). Thus, while YxxP phosphorylation activates Cas, YDYV phosphorylation provides negative feedback, restraining leading edge dynamics and stabilizing adhesions.

Cas receives additional regulatory input from NSP family proteins, including BCAR3 (breast cancer anti-estrogen resistance 3, also called AND-34, NSP2, Sh2d3b) (*Wallez et al., 2012*). Cas forms a complex with BCAR3, with the Cas CCH domain bound to a CDC25H domain in BCAR3 (*Figure 1a*). BCAR3 and Cas cooperate in many biological assays, with each protein requiring the other. For example, BCAR3 increases Cas phosphorylation and Cas-dependent membrane ruffling, adhesion disassembly, cell migration, cell proliferation (*Cai et al., 2003*; *Cross et al., 2016*; *Oh et al., 2013*; *Riggins et al., 2003*; *Roselli et al., 2010*; *Schrecengost et al., 2007*; *Schuh et al., 2010*). Like Cas, BCAR3 protects estrogen-dependent breast cancer cells from inhibitory actions of anti-estrogens (*van Agthoven et al., 1998*). However, it remains unclear how Cas and BCAR3 cooperate. Several lines of evidence suggest that BCAR3 may be regulated by tyrosine phosphorylation. First, it contains an SH2 domain that binds pY proteins, including the epidermal growth factor (EGF) receptor (*Oh et al., 2008*) and PTPRA (RPTPα), a protein phosphatase that activates SFKs and Cas (*Sun et al., 2012*; *von Wichert et al., 2003*; *Zheng et al., 2000*). Second, BCAR3 tyrosine phosphorylation is stimulated by adhesion or serum, although the significance of the phosphorylation is unknown (*Cai et al., 1999*). These studies suggest that BCAR3 interacts with tyrosine kinases and is phosphorylated, but leave open whether BCAR3 phosphorylation is involved in Cas activation.

Here, we report that we detected BCAR3 in a screen for pY proteins that are down-regulated by CRL5 and the proteasome. We identified BCAR3 Y117 as a phosphorylation site that recruits SOCS6

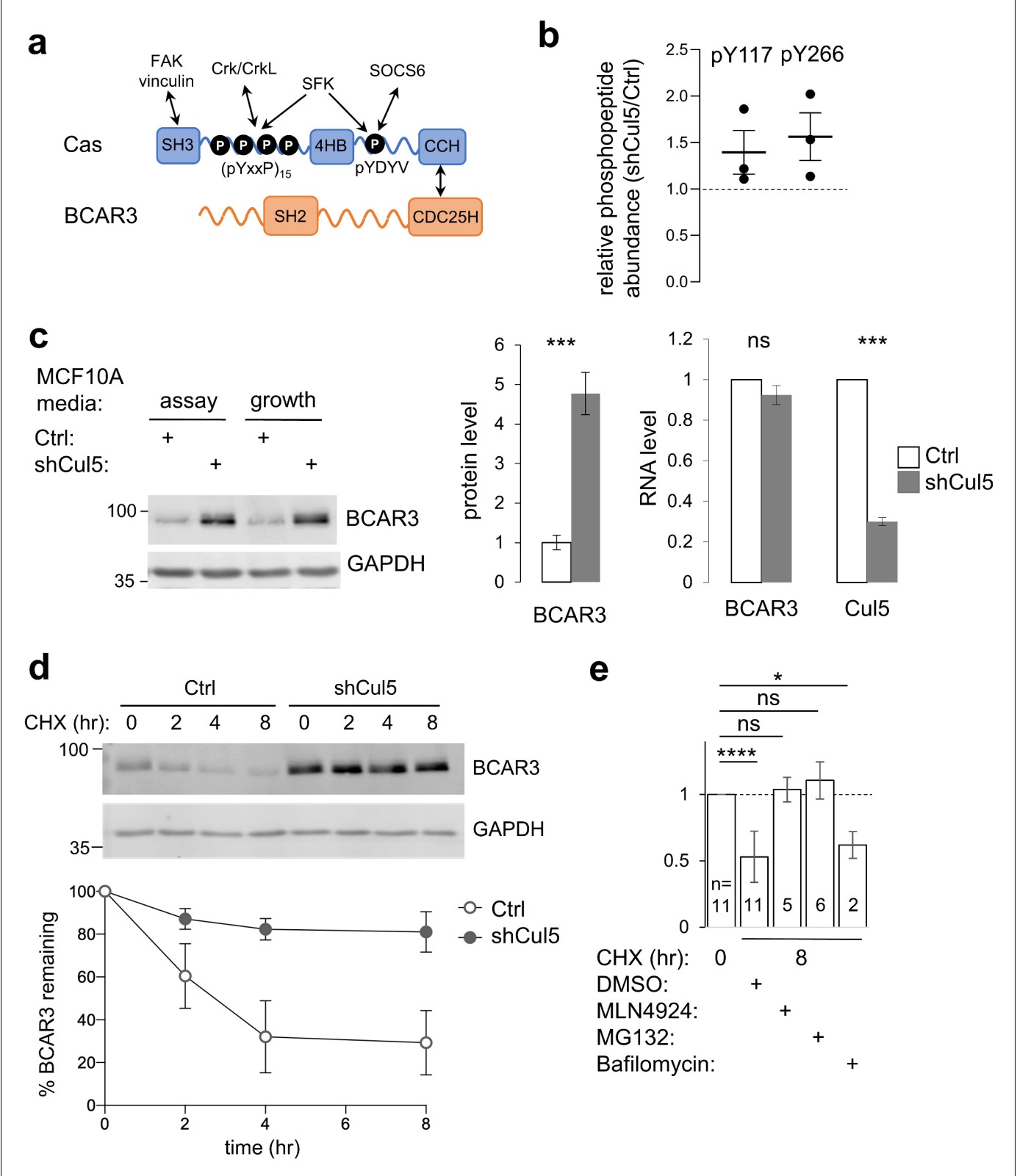

**Figure 1.** CRL5 regulates BCAR3 protein turnover. (**a**) Cas and BCAR3 structures. (**b**) Ratio of BCAR3 pY117 and pY266 phosphopeptide abundance in Cul5-deficient (shCul5) relative to control (Ctrl) MCF10A cells. n=three biological replicates. (**c**) BCAR3 protein and RNA levels in Ctrl and shCul5 cells. Representative immunoblot of cells cultured in assay or growth media (see Methods). Quantification of BCAR3 immunoblots and RNA analysis by qPCR, both normalized to GAPDH. Mean ± SD ; n=3 biological replicates. ***p < 0.001 (t-test). (**d**) BCAR3 degradation. Control and Cul5-deficient cells

*Figure 1 continued*

were treated with cycloheximide (CHX) for various times. Representative immunoblot and quantification. Mean ± SEM ; n=3 biological replicates. (e) Proteasome-dependent BCAR3 degradation. MCF10A cells were treated for 8 hr with CHX and either MLN4924 (cullin neddylation inhibitor), MG132 (proteasome inhibitor), or bafilomycin A1 (lysosome inhibitor). Mean ± SD; the number of biological replicates is noted on the graph. *p < 0.05; ****p < 0.0001 (One-way ANOVA).

and leads to CRL5-dependent degradation. In addition, we found that BCAR3 is needed for single-cell migration and invasion, and for the increased lamellipodial ruffling and collective migration of Cul5-deficient cells. Using gene silencing and mutant analysis, we find that Cas localizes to adhesions independent of BCAR3 but BCAR3 localization to adhesions requires association with Cas. In the adhesion, BCAR3 activates SFKs, Cas phosphorylation, membrane ruffling and cell migration, dependent on both Y117 and the SH2 domain. BCAR3 and Cas thus form a signaling hub that localizes to active integrins and coordinates actin dynamics under negative control by the ubiquitin-proteasome system.

## Results

### CRL5 regulates BCAR3 protein turnover

We previously reported that CRL5 inhibits Src activity and Src-dependent transformation of MCF10A epithelial cells, in part by targeting pY proteins such as pY-Cas for degradation by the ubiquitin-proteasome system (*Teckchandani et al., 2014*). Because overexpression of Cas alone did not phenocopy CRL5 inhibition (*Teckchandani et al., 2014*), we infer that CRL5 down-regulates additional pY proteins that become limiting when Cas is over-expressed. We sought to identify such pY proteins by screening for pY peptides whose abundance increases when Cul5 is inhibited. To this end, control and Cul5-deficient MCF10A cells were lysed under denaturing conditions, proteins were digested with trypsin, and peptides were labeled with isobaric TMT tags for quantitative pY proteomics (*Zhang et al., 2007*). In one experiment, samples were prepared from control and Cul5-deficient cells that were starved for epidermal growth factor (EGF) for 0, 24, or 72 hr. Starvation time had no systematic effect on peptide abundance, so, in a second experiment, we prepared biological triplicate samples from growing control and Cul5-deficient cells. Sixteen pY peptides increased significantly in Cul5-deficient cells in both experiments, including pY128 from Cas and pY117 and pY266 from BCAR3 (*Figure 1b*, *Table 1*, *Supplementary File 1*). We decided to focus on BCAR3 because Cul5 regulates Cas and Cas binds BCAR3 (*Wallez et al., 2012*).

The increased quantity of BCAR3 pY peptides in Cul5-deficient cells could result from increased phosphorylation, increased protein level, or both. We used immunoblotting to test whether BCAR3 protein level is regulated by Cul5. BCAR3 protein level increased approximately fourfold in Cul5-deficient cells, under two different media conditions (*Figure 1c*). However, RNA levels were unaltered (*Figure 1c*), consistent with altered protein synthesis or degradation. To measure degradation, we inhibited new protein synthesis with cycloheximide and monitored BCAR3 protein level. BCAR3 half-life was approximately 4 hr in control cells but greater than 8 hr in Cul5-deficient cells (*Figure 1d*). BCAR3 degradation was inhibited by Cullin neddylation inhibitor MLN4924 or proteasome inhibitor MG132, but not by lysosome inhibitor Bafilomycin (*Figure 1e*). These results suggest that CRL5 regulates BCAR3 turnover by the ubiquitin-proteasome system and that the increase in BCAR3 pY117 and pY266 in Cul5-deficient cells is likely due to an increased availability of BCAR3 protein for phosphorylation rather than, or in addition to, an increase in kinase activity.

### BCAR3 regulates epithelial cell migration

BCAR3 is required for the migration of cancer cells and fibroblasts in single-cell assays (*Cross et al., 2016*; *Schrecengost et al., 2007*; *Sun et al., 2012*) but its importance in single and collective epithelial cell migration is unknown. We inhibited BCAR3 expression in MCF10A cells using siRNA or *BCAR3* gene disruption (*Figure 2a*, *Figure 2—figure supplement 1a*). BCAR3-deficient cells migrated slower than control cells in single-cell migration and invasion assays, regardless of Cul5, suggesting that BCAR3 and CRL5 regulate single-cell migration independently (*Figure 2b,c*, *Figure 2—figure supplement 1b and c*). In contrast, in a collective migration scratch wound assay,

**Table 1.** Phosphotyrosine peptides increased in Cul5-deficient cells.

Protein names, tyrosine positions, Uniprot accession numbers, peptide sequences, and quantification of phosphotyrosine peptides that were significantly increased in two independent experiments, each performed in triplicate, comparing Cul5-deficient and control MCF10A cells. In Experiment 1, the triplicate samples were obtained from cells starved for EGF for 0, 24, or 72 hr. In Experiment 2, the triplicate samples were all from unstarved cells. [a] Ratio shCul5/Ctrl. [b] p value, t-test (two-tailed, paired); n=3.

| Protein | Position | Uniprot | Sequence | Experiment 1 | | Experiment 2 | |
|---------|----------|---------|----------|--------------|------|--------------|------|
| | | | | Fold[a] | p[b] | Fold[a] | p[b] |
| KRT6A | Y62 | Q92625 | sLyGLGGSk | 3.21 | 0.012 | 1.85 | 0.011 |
| ABL1 | Y393 | P00519 | lMTGDTyTAHAGAk | 2.90 | 0.010 | 1.77 | 0.001 |
| ANKS1A | Y455 | Q92625 | eEDEHPyELLLTAETk | 2.58 | 0.004 | 1.41 | 0.009 |
| ARHGAP35 | Y1105 | Q9NRY4 | nEEENIYsVPHDSTQGk | 2.49 | 0.008 | 1.67 | 0.005 |
| RIN1 | Y36 | Q13671 | ekPAQDPLyDVPNASGGQAGGPQRPGR | 2.02 | 0.018 | 1.45 | 0.018 |
| BCAR3 | Y117 | O75815 | dPHLLDPTVEyVk | 1.96 | 0.011 | 1.34 | 0.049 |
| MPZL1 | Y263 | O95297 | sESVVyADIR | 1.90 | 0.026 | 1.54 | 0.027 |
| BCAR3 | Y266 | O75815 | cLEEHyGTSPGQAR | 1.81 | 0.000 | 1.50 | 0.012 |
| BCAR1 | Y128 | P56945 | aQQGLyQVPGPSPQFQSPPAk | 1.80 | 0.003 | 1.28 | 0.036 |
| SGK223 | Y413 | Q86YV5 | eATQPEPIyAESTk | 1.74 | 0.024 | 1.26 | 0.032 |
| ENO1 | Y44 | P06733 | aAVPSGASTGIyEALELR | 1.64 | 0.004 | 1.18 | 0.032 |
| ITGB4 | Y1207 | P16144 | vcAYGAQGEGPySSLVScR | 1.59 | 0.010 | 1.19 | 0.043 |
| ANXA2 | Y30 | A6NMY6 | ayTNFDAER | 1.58 | 0.018 | 1.31 | 0.014 |
| IGF1R | Y1165 | P08069 | dIYETDyYR | 1.53 | 0.030 | 1.69 | 0.001 |
| PTPRA | Y798 | P18433 | vVQEYIDAFSDyANFk | 1.50 | 0.006 | 1.23 | 0.019 |
| TLN1 | Y26 | Q9Y480 | tMQFEPSTMVyDAcR | 1.22 | 0.022 | 1.25 | 0.014 |

BCAR3 was not required unless Cul5 was depleted (*Figure 2d*). Moreover, inspection of the wound edge revealed that BCAR3 is also needed for the increased lamellipodia length and ruffling in Cul5-depleted cells (*Figure 2e–g*). This epistatic relationship is consistent with CRL5 inhibiting BCAR3-dependent migration and lamellipodia under collective conditions, as found before for Cas (*Teckchandani et al., 2014*). We do not understand the differences between single-cell and collective migration, but can make use of single-cell assays to test the role of BCAR3 in normal cells and collective assays to test the role of BCAR3 when it is over-expressed or activated by Cul5 depletion.

## CRL5 directly targets BCAR3 through SOCS6

CRL5 promotes ubiquitination and degradation of substrate proteins bound to adaptor proteins (*Okumura et al., 2016*). CRL5 adaptors include SOCS family proteins, which contain SH2 domains for binding to pY. To test whether SOCS proteins bind BCAR3, we transiently expressed T7-tagged SOCS proteins and assayed binding to endogenous BCAR3 by immunoprecipitation and immuno-blotting. BCAR3 specifically co-precipitated with SOCS6, the same adaptor that binds Cas (*Figure 3a*; *Teckchandani et al., 2014*). Accordingly, SOCS6 depletion increased BCAR3 steady-state protein levels and decreased the rate of BCAR3 turnover (*Figure 3b,c*). Together, these results suggest that CRL5[SOCS6] mediates BCAR3 turnover.

Since BCAR3 and Cas bind each other and both are bound and regulated by SOCS6, it is possible that SOCS6 binds BCAR3 indirectly, through Cas. We tested this possibility by two strategies (*Figure 3d*). First, we used a BCAR3 mutant, L744E/R748E, called here BCAR3[EE], that does not bind Cas (*Wallez et al., 2014*). This mutation inhibited binding to Cas but not SOCS6 (*Figure 3e*). Second, we found that BCAR3-SOCS6 binding occurred in cells from which Cas had been depleted with siRNA (*Figure 3f*). Collectively, these data suggest that SOCS6 binds to BCAR3 independently of Cas.

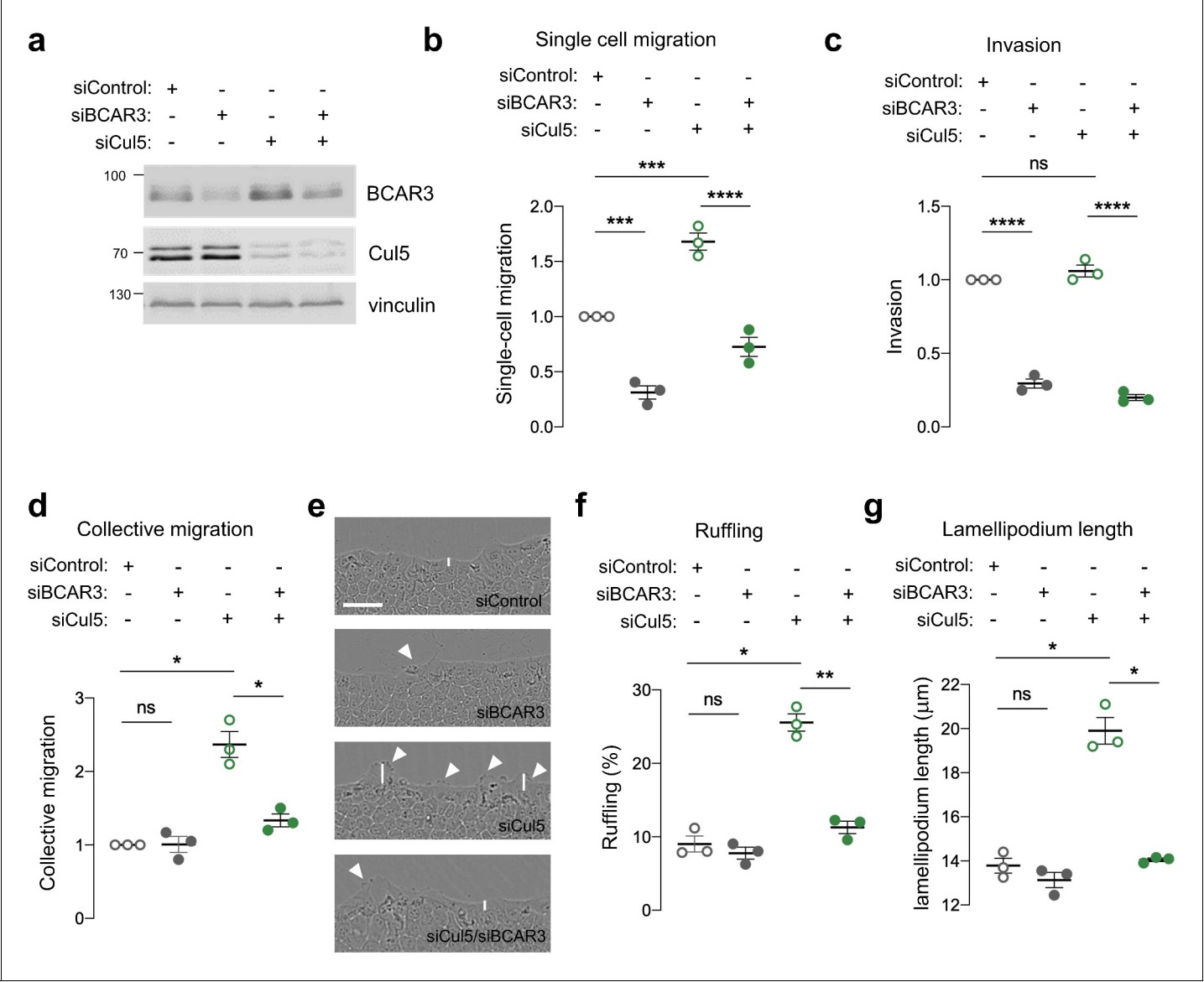

**Figure 2.** BCAR3 regulates epithelial cell migration. MCF10A cells were transfected with control, BCAR3, or Cul5 siRNA. (**a**) Representative immunoblot showing BCAR3, Cul5, and vinculin protein levels. (**b**) Single cell migration using Boyden chamber assay. Mean ± SEM; n=3 biological replicates, each with five technical replicates. ***p<0.0005 and ****p<0.0001 (One-way ANOVA). (**c**) Invasion using Boyden chamber containing Matrigel. Mean ± SEM; n=3 biological replicates, each with five technical replicates. ****p<0.0001 (One-way ANOVA). (**d–g**) Collective migration. Confluent monolayers were placed in assay media and wounded. (**d**) Relative migration after 12 hr. Mean ± SEM; n=3 biological replicates each with 8–12 technical replicates. *p<0.05 (One-way ANOVA). (**e**) Representative images of scratch wounds after 6 hr of migration. Arrows indicate cells with membrane ruffles and lines indicate lamellipodia length measurements. Scale bar: 100 μm. (**f**) Percentage of ruffling cells. Mean ± SEM of >250 cells per condition from n=3 biological replicates. *p<0.05 and **p<0.005 (One-way ANOVA). (**g**) Lamellipodia length. Mean ± SEM of 50 cells per condition from n=3 biological replicates. *p<0.05 (One-way ANOVA).

The online version of this article includes the following figure supplement(s) for figure 2:

**Figure supplement 1.** *BCAR3* gene deletion inhibits single-cell migration and invasion.

## SOCS6 binds BCAR3 pY117

We considered that SOCS6 might bind BCAR3 through pY-dependent or -independent interactions. Pervanadate, a cell-permeable phosphotyrosine phosphatase inhibitor, increased BCAR3-SOCS6 association, suggesting pY dependence (*Figure 4a*). In addition, disrupting the SOCS6 SH2 domain by deletion (ΔC) or point mutation (R407K) inhibited binding to BCAR3 (*Figure 4b*). This suggests

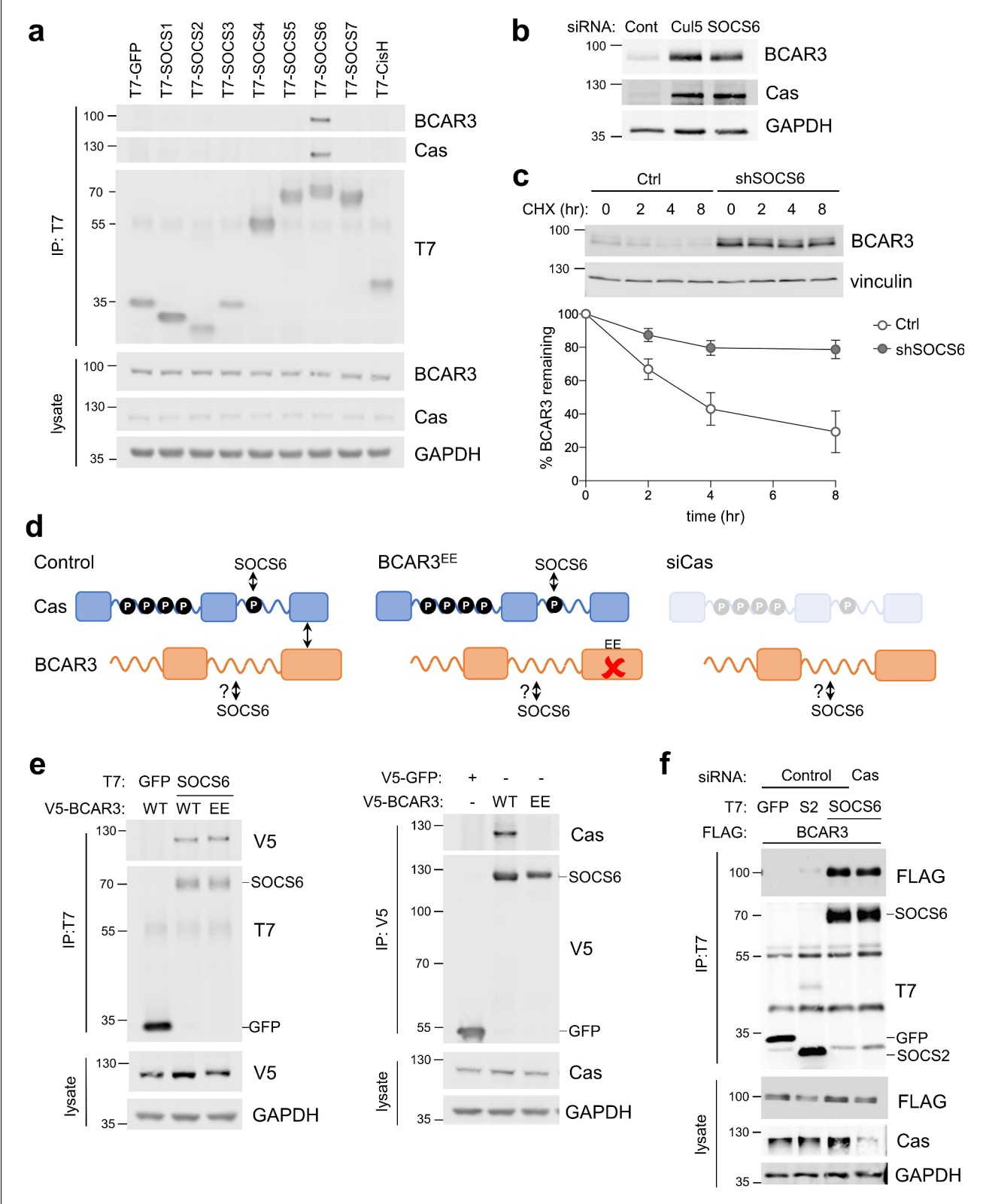

**Figure 3.** SOCS6 regulates BCAR3 stability independently from Cas.  (**a**) SOCS6 binds BCAR3 and Cas. HeLa cells were transfected with T7-tagged GFP or SOCS proteins and treated with pervanadate for 30 min before lysis. Lysates were immunoprecipitated with T7 antibody and immunoprecipitates and lysates were immunoblotted with antibodies to BCAR3, Cas and T7. (**b**) SOCS6 regulates BCAR3 protein level. MCF10A cells were transfected with control, Cul5, or SOCS6 siRNA and analyzed by immunoblotting. (**c**) BCAR3 degradation. MCF10A control and SOCS6-deficient

*Figure 3 continued on next page*

*Figure 3 continued*

cells were treated with cycloheximide (CHX) for various times. Representative immunoblot and quantification. Mean ± SEM; n=3 biological replicates. (d) Strategy for testing whether SOCS6 binding to BCAR3 requires Cas. (e) BCAR3 LxxE/RxxE mutation inhibits binding to Cas but not SOCS6. Left panel: HeLa cells were transfected with control vector or T7-SOCS6 and SNAP-V5-BCAR3$^{WT}$ or SNAP-V5-BCAR3$^{EE}$. Right panel: HeLa cells were transfected with control vector, SNAP-V5-BCAR3$^{WT}$ or SNAP-V5-BCAR3$^{EE}$. Lysis, immunoprecipitation and immunoblot as in (a). (f) SOCS6 binds BCAR3 in Cas-deficient cells. HeLa cells were treated with control or Cas siRNA and transfected with T7-SOCS6 and 3xFLAG-BCAR3. Lysis, immunoprecipitation and immunoblot as in (a).

that the SOCS6 SH2 domain binds pY-BCAR3. Serum or adhesion stimulates tyrosine phosphorylation of BCAR3 in mouse fibroblasts (*Cai et al., 1999*), but the specific sites have not been identified. High-throughput pY proteomics surveys have identified phosphorylation of BCAR3 at tyrosine residues 42, 117, 212, 266, and 429 in over 25 mouse and human cell lines (*Hornbeck et al., 2019*). We tested whether these sites were required for binding SOCS6 using site-directed mutagenesis (*Figure 4c*). BCAR3$^{F5}$, in which all five tyrosines were changed to phenylalanines, was unable to bind SOCS6 (*Figure 4d*). By mutating each pY site individually, we found that Y117 is necessary to bind SOCS6 (*Figure 4d*). Furthermore, mutating all sites except Y117 (42, 212, 266, and 429) had little effect on SOCS6 binding (BCAR3$^{F4}$ mutant, *Figure 4e*). These results support a model in which phosphorylation of BCAR3 at Y117 is both necessary and sufficient to bind the SOCS6 SH2 domain.

## CRL5-dependent BCAR3 turnover requires Y117 and Cas association, but not the SH2 domain or other tyrosine residues

To test whether Y117 or other domains of BCAR3 are required for CRL5-dependent BCAR3 protein turnover, we measured the effect of various mutations on the level of tagged BCAR3 protein in control and Cul5-deficient cells. To avoid possible artifacts due to over-expression, we used a doxycycline-inducible promoter (*Baron et al., 1997*). MCF10A cells were first transduced to express the reverse *tet* transactivator (rtTA), and then transduced to express SNAP-V5-tagged wildtype or mutant BCAR3 under control of the *tet* operator. Cells were treated with doxycycline (dox) to induce wildtype or mutant BCAR3 expression, with or without knocking down endogenous BCAR3 with an siRNA targeting the 3′ UTR.

We first examined the role of Y117 in BCAR3 turnover. BCAR3$^{Y117F}$ was expressed at approximately twofold higher level than BCAR3$^{WT}$ at the same concentration of dox (*Figure 5a*). Moreover, depleting Cul5 increased the level of BCAR3$^{WT}$ more than twofold while the level of BCAR3$^{Y117F}$ was unchanged (*Figure 5b*). This suggests that CRL5 regulates BCAR3 protein level dependent on Y117. BCAR3$^{F4}$, which contains Y117 but not four other tyrosine phosphorylation sites, was also regulated by CRL5 (*Figure 5c*). These results are consistent with SOCS6 binding to pY117 and targeting BCAR3 for CRL5-dependent degradation.

We extended this approach to test the importance of the BCAR3 SH2 domain and Cas binding site for degradation. To inactivate the BCAR3 SH2 domain, we created an arginine to lysine at residue 177 (R177K), which lies in the consensus FLVRES motif and is required to bind the pY phosphate (*Jaber Chehayeb and Boggon, 2020*; *Marengere and Pawson, 1994*). Cul5-depletion increased the level of BCAR3$^{R177K}$ (*Figure 5d*), suggesting this mutant is still phosphorylated at Y117 and targeted by CRL5. In contrast, BCAR3$^{EE}$, which binds SOCS6 (*Figure 3e*) but not Cas (*Figure 3e* and *Figure 5—figure supplement 1*), was not regulated by CRL5. Taken together, these results suggest that pY117 and Cas association are required for CRL5-dependent turnover of BCAR3 expressed at near endogenous level.

## Lamellipodial ruffling and cell migration require BCAR3 Y117, SH2 domain, and Cas association

Over-expression of BCAR3 in fibroblasts and breast cancer cells stimulates Cas-dependent functions, such as lamellipodia ruffling (*Cai et al., 2003*; *Wallez et al., 2014*; *Wilson et al., 2013*). Similarly, BCAR3$^{WT}$ increased membrane ruffling when over-expressed in MCF10A cells (*Figure 6a*). Since BCAR3$^{Y117F}$ accumulates to higher levels, we suspected it may be more active in biological assays. However, contrary to expectations, ruffling was induced by over-expressing BCAR3$^{F4}$ but not BCAR3$^{Y117F}$, BCAR3$^{R177K}$ or BCAR3$^{EE}$ (*Figure 6a*). Similarly, when endogenous BCAR3 is depleted, BCAR3$^{WT}$ and BCAR3$^{F4}$, but not BCAR3$^{Y117F}$, BCAR3$^{R177K}$ or BCAR3$^{EE}$, were able to rescue single-

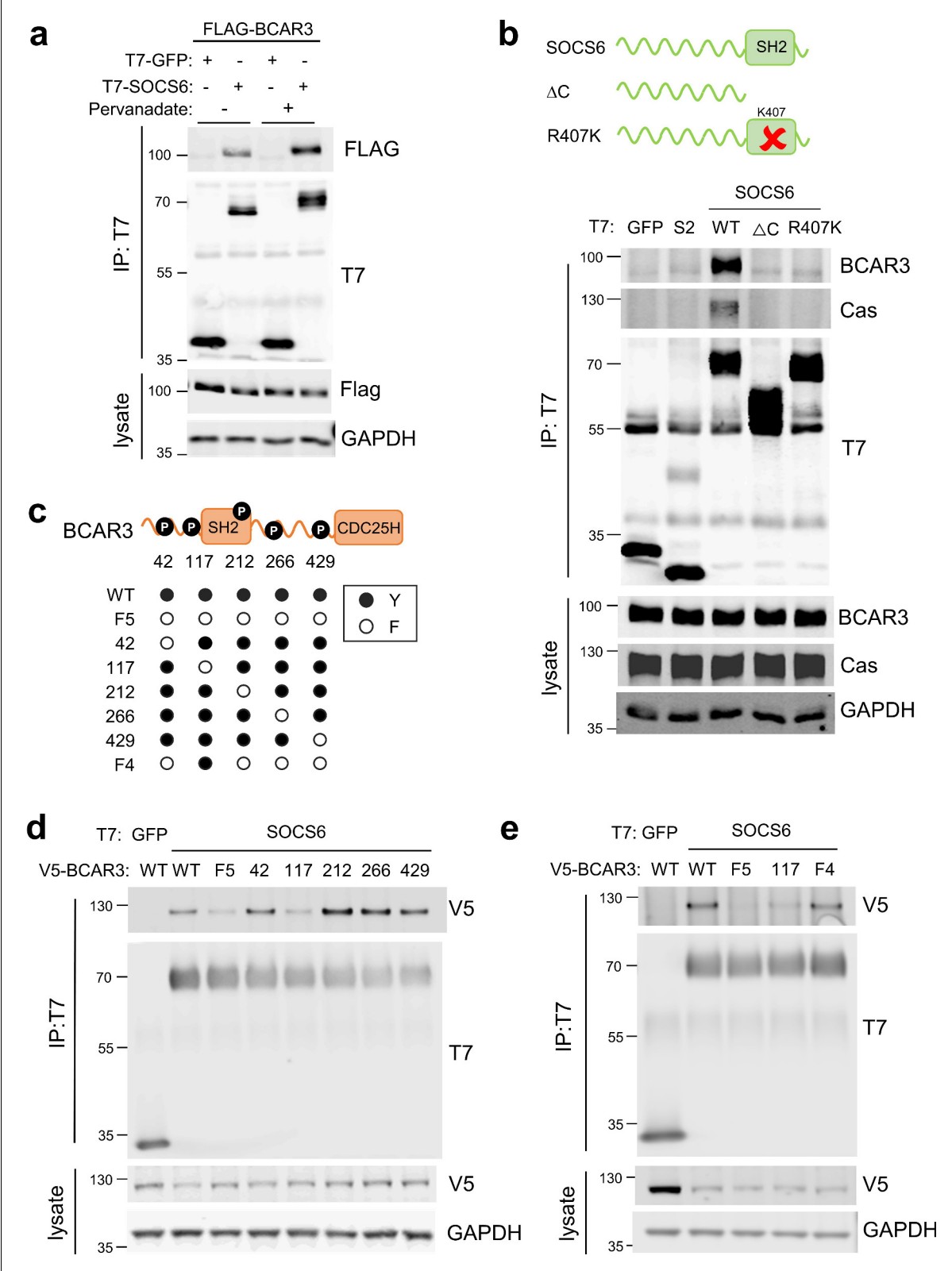

**Figure 4.** SOCS6 binds BCAR3 pY117. (a) Phosphatase inhibition increases SOCS6-BCAR3 binding. HeLa cells were transfected with T7-GFP or T7-SOCS6 and 3xFLAG-BCAR3 and treated with pervanadate or vehicle. Lysates were immunoprecipitated with T7 antibody and immunoprecipitates and lysates were immunoblotted with T7 and FLAG antibodies. (b) SOCS6-BCAR3 binding requires the SOCS6 SH2 domain. Cells were transfected with control vector and T7-tagged SOCS constructs, treated with pervanadate and lysed. Immunoprecipitation and immunoblotting as in (a). (c) BCAR3$^{WT}$

*Figure 4 continued on next page*

Figure 4 continued

and phenylalanine (F) mutants. Residues Y42, Y117, Y212, Y266, and Y429 phosphorylation sites have been detected across a range of cell lines (http://www.PhosphoSite.org). (d, e) BCAR3 Y117 is necessary and sufficient for SOCS6 binding. HeLa cells were transfected with T7-GFP or T7-SOCS6 and SNAP-V5-BCAR3$^{WT}$ or YF mutants. Transfected cells were treated with pervanadate and lysed. Immunoprecipitation and immunoblotting as in (a).

cell migration (*Figure 6b*) and increase collective migration and lamellipodia ruffling of Cul5-deficient cells (*Figure 6c*). This suggests that Y117 has two roles. First, Y117 is the main phosphorylation site for BCAR3 down-regulation. Second, it is also the main phosphorylation site for BCAR3 function, cooperating with the SH2 domain and Cas to promote single-cell and collective migration and lamellipodial dynamics in the presence and absence of Cul5.

## Cas recruits BCAR3 to integrin adhesions

These findings raise the question of how BCAR3 Y117, SH2 domain, and Cas binding cooperate to regulate MCF10A cell motility. Previous studies in cancer cells and fibroblasts found that BCAR3 localizes to integrin adhesions (*Cross et al., 2016*; *Sun et al., 2012*) and that BCAR3 over-expression can increase Cas in membrane ruffles (*Riggins et al., 2003*). Increased Cas could then activate lamellipodia dynamics through the established Cas/Crk/DOCK180/Rac pathway (*Klemke et al., 1998*; *Sakai et al., 1994*; *Sanders and Basson, 2005*; *Sharma and Mayer, 2008*). These findings suggest that BCAR3's Y117 and SH2 domain may be needed for BCAR3 to correctly localize Cas. To explore this possibility, we monitored the effect of depleting BCAR3 on Cas localization and the effect of depleting Cas on BCAR3 localization. We were unable to detect endogenous BCAR3 by immunofluorescence with available antibodies, so we expressed SNAP-V5-BCAR3 at near endogenous levels and detected the fusion protein with SNAP ligand (*Grimm et al., 2015*; *Keppler et al., 2003*). Both Cas and BCAR3$^{WT}$ localized to adhesions in the leading edge of collectively migrating cells. Cas remained in adhesions when BCAR3 was depleted (*Figure 7a*). In contrast, BCAR3 was absent from adhesions when Cas was depleted (*Figure 7b*). Thus, BCAR3 requires Cas to localize in adhesions, and not vice versa. Consistently, all BCAR3 mutants that bound Cas were present in adhesions (*Figure 8a,b*, *Figure 5—figure supplement 1*). Together, these results suggest that Cas localizes to adhesions independent of BCAR3 and that Cas brings BCAR3 to adhesions by direct binding.

## BCAR3 activates Cas, dependent on BCAR3 Y117 and SH2 domain

Since BCAR3 does not regulate Cas localization, what is the function of Y117 and the SH2 domain? We considered that BCAR3 may activate Cas. To monitor Cas activity, we used antibodies to pY165, one of the repeated YxxP motifs in Cas that recruit Crk (*Fonseca et al., 2004*; *Sakai et al., 1994*; *Songyang et al., 1993*). Cas pY165 in leading edge adhesions was abolished by SFK inhibition or Cas depletion, consistent with Cas activity (*Figure 8—figure supplement 1*). Notably, Cas activity was inhibited when BCAR3 was depleted (*Figure 8—figure supplement 2a*). As expected, BCAR3$^{WT}$ and BCAR3$^{F4}$ rescued Cas activity in BCAR3-depleted cells (*Figure 8a,c*). However, Cas activity was not rescued by BCAR3$^{Y117F}$, BCAR3$^{R177K}$, or BCAR3$^{EE}$, even though Cas was still present (*Figure 8a,c*, *Figure 8—figure supplement 3*). Cas activation in adhesions correlated with the rescue of ruffling and migration (*Figures 6b* and *8*). This suggests that BCAR3 not only has to be bound to Cas but also needs Y117 and its SH2 domain to activate Cas and promote downstream signaling. To test whether BCAR3 activates Cas specifically or phosphorylation more generally, we also stained for FAK pY397 and pY861. The former is a FAK autophosphorylation site and the latter is phosphorylated by SFKs (*Mitra and Schlaepfer, 2006*). We found that BCAR3 depletion did not inhibit FAK autophosphorylation at pY397 but did inhibit FAK pY861 (*Figure 8—figure supplement 2b and c*), consistent with a general role of BCAR3 in stimulating SFKs in adhesions. Taken together, these results suggest that BCAR3 is brought to integrin adhesions by binding to Cas, where it uses its Y117 and SH2 domain to activate SFKs, Cas and downstream signaling, leading to lamellipodial ruffling and migration.

## Discussion

Cell migration requires biophysical and biochemical signaling between the leading edge and adhesion complexes to coordinate adhesion dynamics with actin polymerization, anchorage of stress

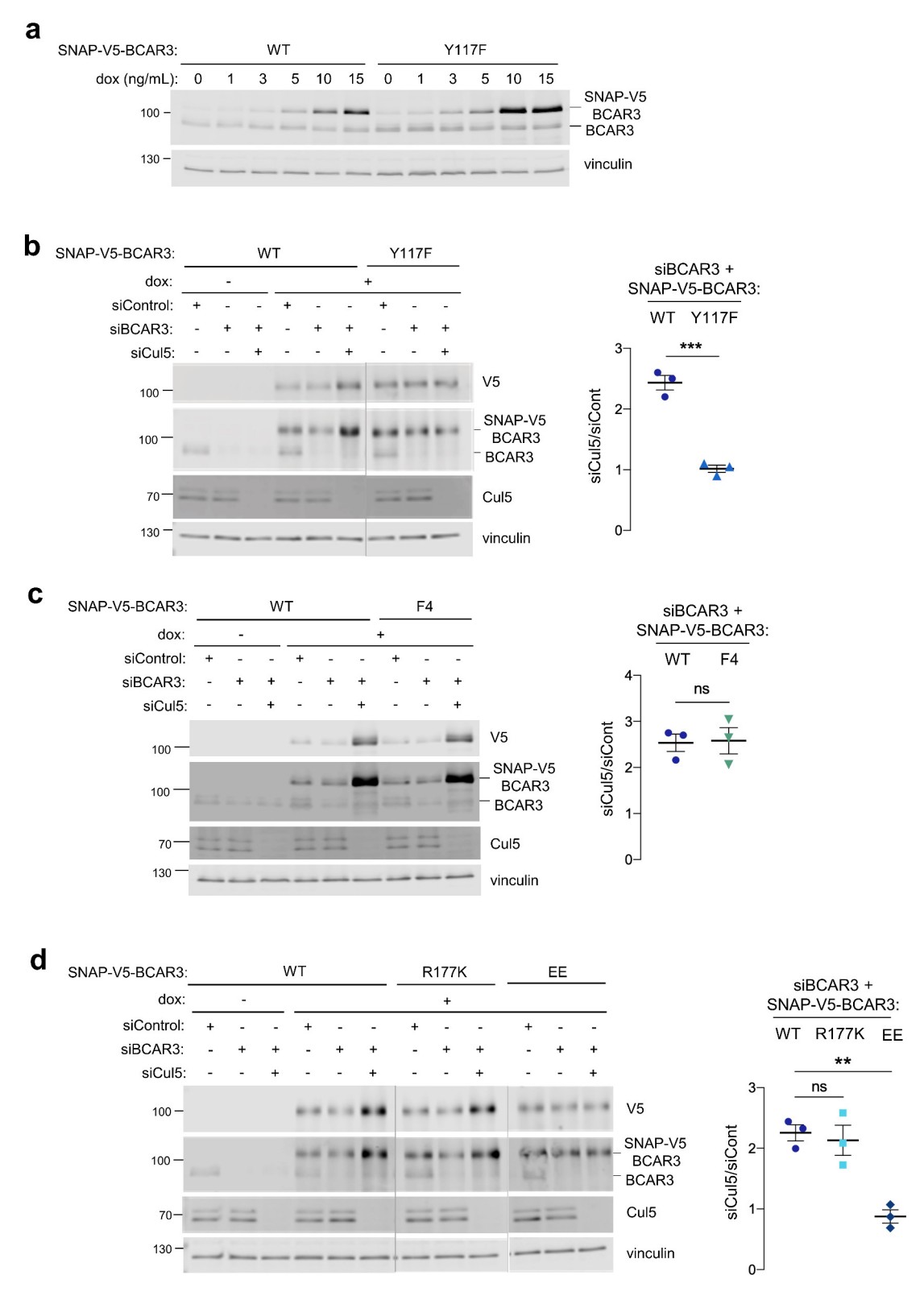

**Figure 5.** Cul5 requires Y117 and Cas association to regulate BCAR3 protein level. (**a**) Y117F mutation increases BCAR3 protein level. SNAP-V5-BCAR3$^{WT}$ and $^{Y117F}$ MCF10A cells were induced with various dox concentrations (ng/mL). Lysates were immunoblotted with BCAR3 antibody to detect endogenous and induced SNAP-V5-BCAR3. (**b**) Cul5 regulates BCAR3 protein level through Y117. Representative immunoblot and quantification of BCAR3 protein in siRNA-treated, dox-induced SNAP-V5-BCAR3$^{WT}$ and $^{Y117F}$ cells. Mean ± SEM; n=3 biological replicates. ***p<0.001 by t-test. (**c**) Cul5

*Figure 5 continued on next page*

*Figure 5 continued*

does not regulate SNAP-V5-BCAR3 $^{F4}$. Mean ± SEM; n=3. ns, not significant. (**d**) Cul5 regulates SNAP-V5-BCAR3$^{R177K}$ but not $^{EE}$. Mean ± SEM; n=3.
**p<0.005 by t-test. Vertical lines indicate different immunoblots, each run with its own wild-type control.
The online version of this article includes the following figure supplement(s) for figure 5:

**Figure supplement 1.** Cas binding to BCAR3 mutants.

---

fibers and generation of traction forces (*Trepat et al., 2012*). Our results address one aspect of this complicated process: the linkage between integrin engagement and lamellipodial extension. Previous studies have found that Cas and BCAR3 synergize for lamellipodium ruffing and cell migration, but they also synergized for protein expression, making the mechanism unclear (*Cai et al., 1999*; *Cross et al., 2016*; *Riggins et al., 2003*; *Schrecengost et al., 2007*; *Wallez et al., 2014*). Here, using loss-of-function approaches, we find that there is positive feedback between Cas and BCAR3 phosphorylation, and dual negative feedback on both proteins by the ubiquitin-proteasome system.

Our results support a multi-step model (*Figure 9*). First, Cas associates with active integrin adhesion complexes near the leading edge of migrating cells through localization signals in its N and C termini. Next, BCAR3 is recruited to adhesions by Cas. Since most BCAR3 in the cell is complexed with Cas (*Cross et al., 2016*), the two proteins are likely recruited jointly, via localization signals on Cas. BCAR3 then stimulates SFKs and Cas phosphorylation in a mechanism that requires BCAR3 SH2 and Y117 but not four other phosphorylation sites. Cas phosphorylation then drives membrane ruffling, presumably utilizing the well-documented mechanisms involving Crk, DOCK180, and Rac

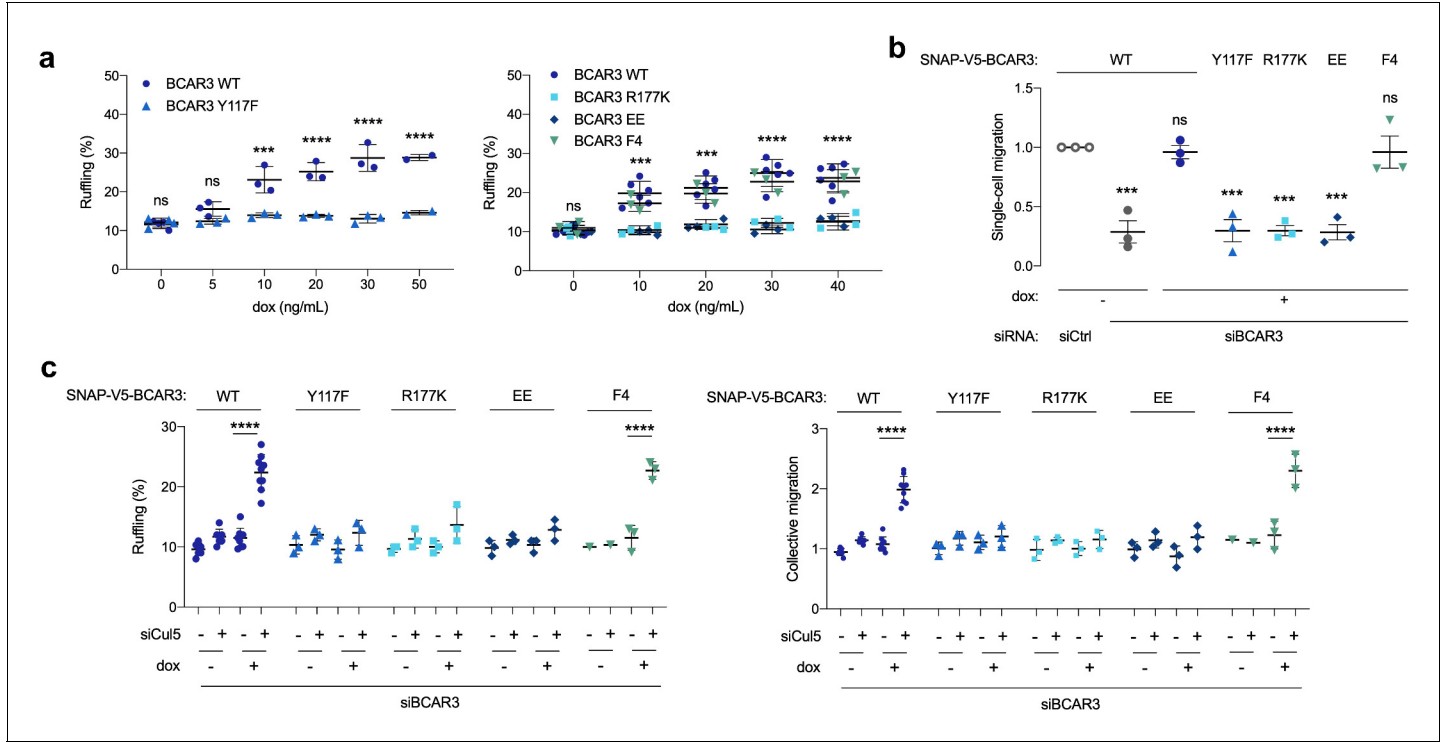

**Figure 6.** BCAR3 Y117, R177 and Cas binding regulate cell migration. (**a**) Over-expression induced ruffling. MCF10A cells transduced with wildtype or mutant SNAP-V5-BCAR3 viruses were induced with various dox concentrations (ng/mL). Confluent monolayers were starved for EGF and wounded. The percentage of ruffling cells was calculated after 6 hr. Mean ± SEM; n=3–6. ***p=0.0001 and ****p<0.0001 (One-way ANOVA). (**b**) Rescue of single-cell migration. Cells were treated with control or BCAR3 siRNA and expression of wildtype or mutant SNAP-V5-BCAR3 induced with 10 ng/mL dox. Boyden chamber assay. Mean ± SEM; n=3 biological replicates, each with five technical replicates. ***p<0.0005 (One-way ANOVA). (**c**) Rescue of Cul5-regulated ruffling and collective cell migration. Cells were treated with BCAR3 siRNA and control or Cul5 siRNA and expression of wildtype or mutant SNAP-V5-BCAR3 induced with 10 ng/mL dox. Scratch wound assay. (Left) Percentage of ruffling cells. Mean ± SEM of >250 cells per condition from n=3–6 experiments. ****p<0.0001 (One-way ANOVA). (Right) Relative migration. Mean ± SEM; n=3–6 biological replicates, each with 8–12 technical replicates. ****p<0.0001 (One-way ANOVA).

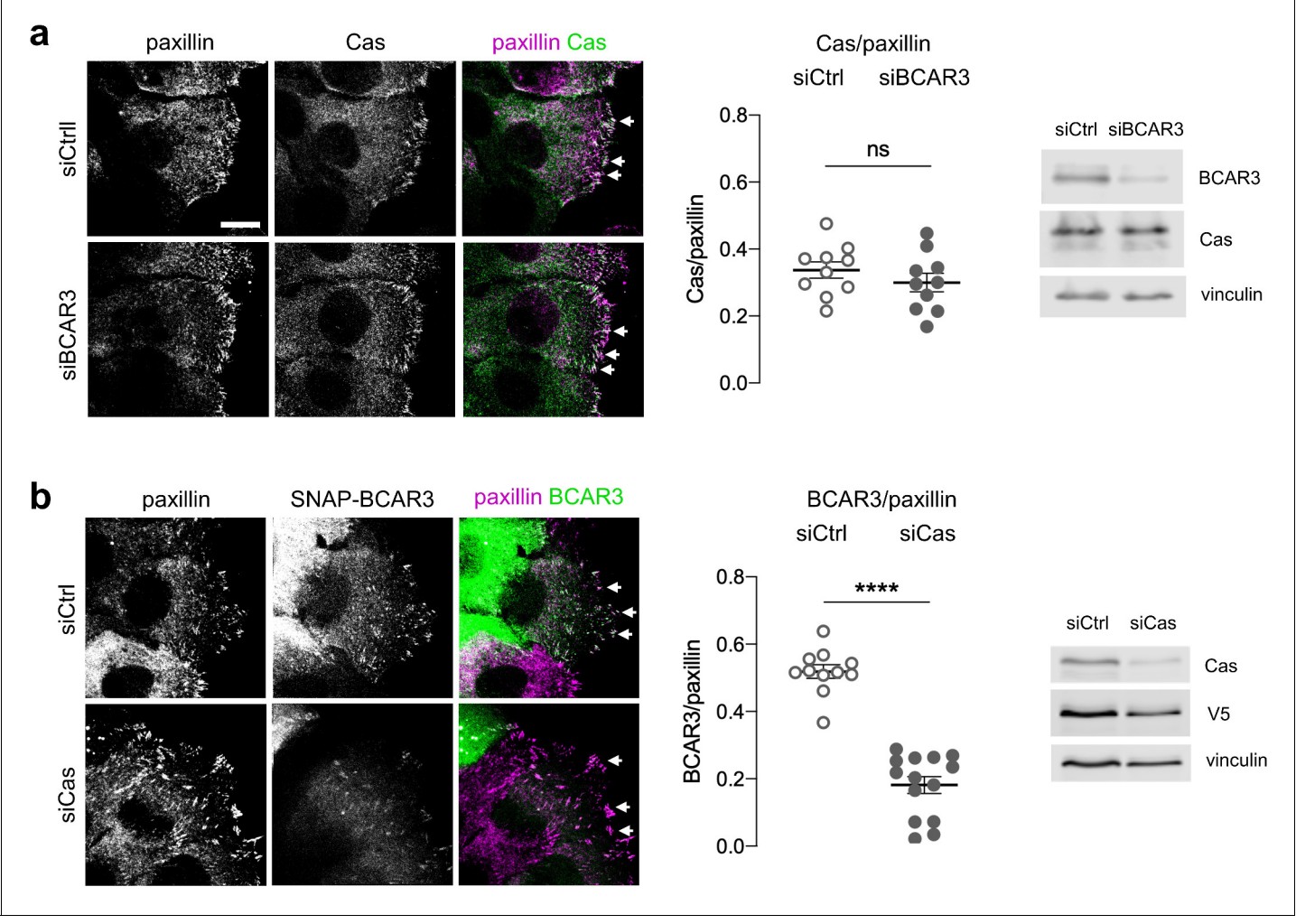

**Figure 7.** Cas recruits BCAR3 to adhesions. (a) Cas localization does not require BCAR3. MCF10A cells were treated with control or BCAR3 siRNA. Scratch wounds were stained for Cas and paxillin. (Left) Maximum intensity projection images. Arrowheads: adhesion sites at the leading edge. Scale bar: 10 µm. (Center) Mean fluorescence intensity of Cas relative to paxillin in adhesion sites at the leading edge. Mean ± SEM; n=10 cells from two biologically independent experiments. ns, not significant. (Right) Immunoblot. (b) BCAR3 localization requires Cas. SNAP-V5-BCAR3[WT] MCF10A cells were treated with control or Cas siRNA and induced with 10 ng/mL dox. Scratch wounds were stained for SNAP and paxillin. (Left) Maximum intensity projection images. (Center) Mean fluorescence intensity of SNAP-BCAR3 relative to paxillin in adhesion sites at the leading edge. Mean ± SEM; n=11–14 cells from two biologically independent experiments. ****p<0.0001 by unpaired t-test. (Right) Immunoblot.

(*Cheresh et al., 1999*; *Klemke et al., 1998*; *Sakai et al., 1994*; *Sanders and Basson, 2005*; *Schaller and Parsons, 1995*; *Sharma and Mayer, 2008*). Thus, the BCAR3 Y117 and SH2 provide a critical switch to activate Cas. Our results also suggest that phosphorylation at Y117 promotes SOCS6 binding and targets BCAR3 for ubiquitination and degradation by the CRL5 E3 ubiquitin ligase. pY-Cas is similarly targeted by CRL5[SOCS6] binding to its pYDYV motif (*Teckchandani et al., 2014*). Ubiquitination of Cas likely occurs within the adhesion (*Teckchandani and Cooper, 2016*), and BCAR3 may also be ubiquitinated in the adhesion. Ubiquitination and degradation of either protein is expected to terminate signaling until new Cas and BCAR3 molecules are recruited from the cytosol, providing double insurance against excess activity.

Where is Y117 phosphorylated and how does it activate SFKs? We have been unable to generate a phospho-specific antibody with the sensitivity and specificity to localize pY117 BCAR3 in cells. However, we suspect that phosphorylation may occur in adhesions, because BCAR3[EE], which does not localize to adhesions, is not subject to pY117-dependent turnover by CRL5. After Y117 is phosphorylated, we infer that BCAR3 activates SFKs in adhesions to phosphorylate Cas and FAK. Indeed, previous studies reported SFK activation by BCAR3 (*Riggins et al., 2003*; *Schuh et al., 2010*;

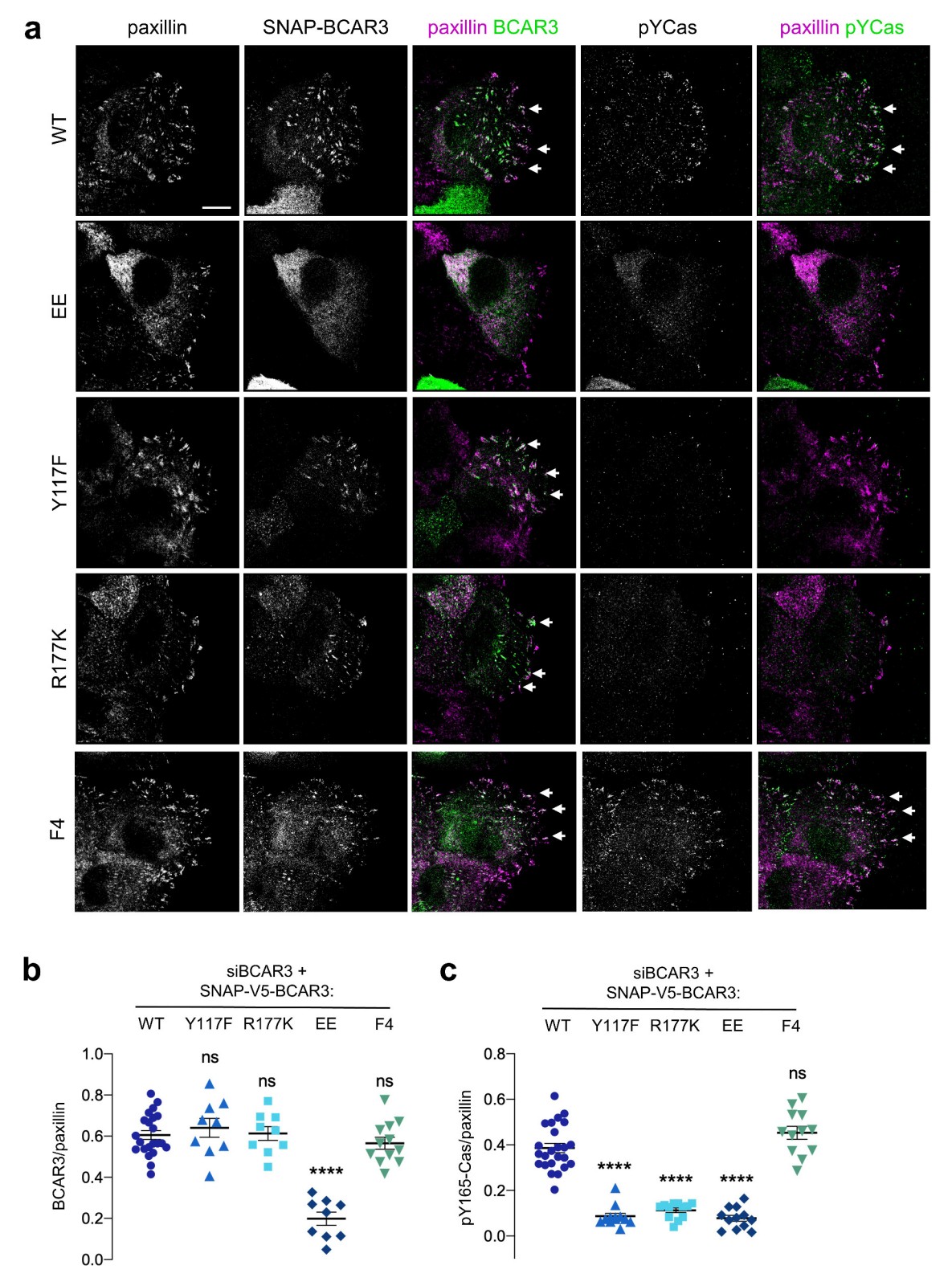

**Figure 8.** BCAR3 stimulates Cas phosphorylation in adhesions, dependent on BCAR3 Y117 and SH2 interactions. (a) Wildtype or mutant SNAP-V5-BCAR3 MCF10A cells were treated with control or BCAR3 siRNA and induced with 10 ng/mL dox. Scratch wound were stained for SNAP, pY165-Cas and paxillin. Maximum intensity projection. Scale bar = 10 μm. (b) Mean fluorescence intensity of SNAP-V5-BCAR3 relative to paxillin in adhesion sites

*Figure 8 continued on next page*

*Figure 8 continued*

at the leading edge. Mean ± SEM. ****p<0.0001 (One-way ANOVA). (**c**) Mean fluorescence intensity of pY165-Cas relative to paxillin in adhesion sites at the leading edge. Mean ± SEM. ****p<0.0001 (One-way ANOVA).

The online version of this article includes the following figure supplement(s) for figure 8:

**Figure supplement 1.** The pY165-Cas antibody is specific for pYCas.

**Figure supplement 2.** BCAR3 is required for pY165-Cas and pY861-FAK, but not pY397-FAK.

**Figure supplement 3.** Cas levels are not altered when BCAR3$^{WT}$ and mutants are expressed.

*Sun et al., 2012*). One potential mechanism involves pY-PTPRA binding to the BCAR3 SH2 domain and bringing the BCAR3-Cas complex to adhesions where Cas is phosphorylated by SFKs (*Sun et al., 2012*). Since PTPRA is able to activate SFKs (*Ponniah et al., 1999*; *Su et al., 1999*; *Zheng et al., 2000*), this is an attractive model. However, in our studies, the BCAR3 SH2 domain is not needed to localize either BCAR3 or Cas, suggesting that Cas-BCAR3 would bring PTPRA to adhesions rather than vice versa. In addition, PTPRA depletion does not inhibit Cas Y165 phosphorylation (data not shown). An alternative is that pY117 binds the SH2 domain of an SFK or an unidentified protein that activates SFKs. The Y117 sequence is conserved across vertebrates and a homologous residue is present in Shep1, a second NSP family protein that activates Cas (*Roselli et al., 2010*). However, the sequence does not suggest which SH2 domain may bind other than SOCS6 (*Krebs et al., 2002*; *Simó and Cooper, 2013*; *Teckchandani et al., 2014*; *Zadjali et al., 2011*; *Table 2*). The presence of a positively charged residue two residues after the phosphosite is predicted to inhibit binding to SFKs (*Songyang et al., 1993*). Therefore, pY117 may bind another SH2 protein that activates SFKs, or may stimulate Cas phosphorylation by an allosteric mechanism or by altering binding to other charged molecules, such as membrane phospholipids.

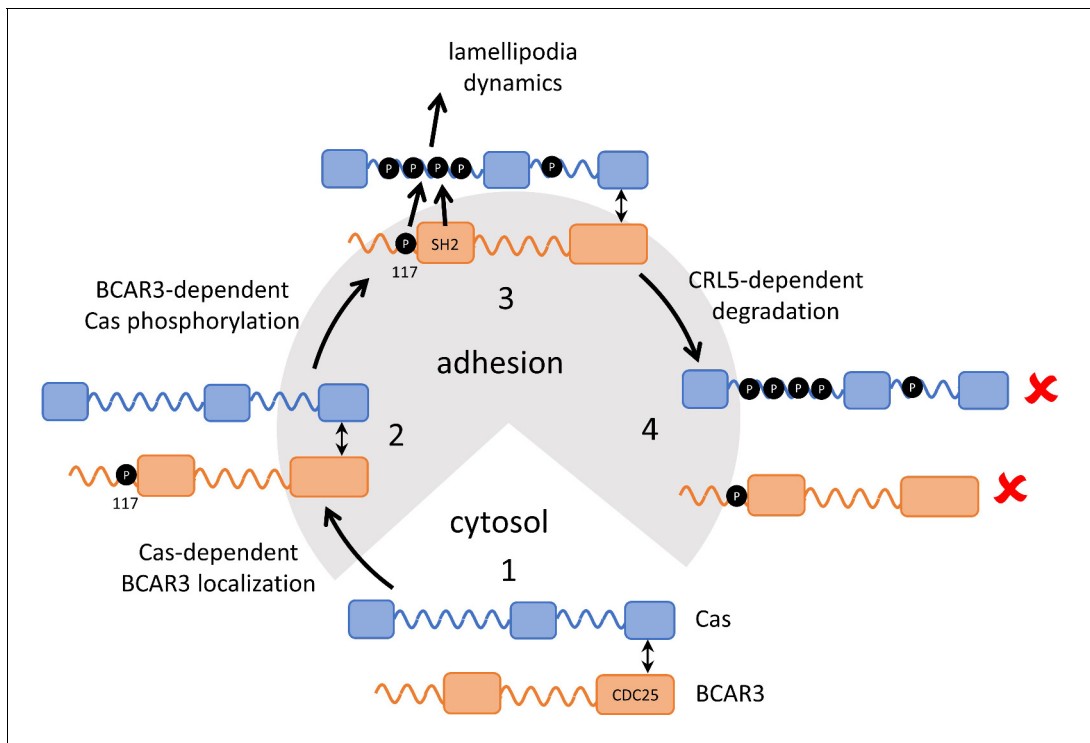

**Figure 9.** Model showing inter-dependence of Cas and BCAR3 for localization and activation. Step 1: A pre-formed BCAR3-Cas complex translocates to integrin adhesions through localization signals in the N and C terminal domains of Cas. Step 2: BCAR3, phosphorylated at Y117 by an unknown kinase, stimulates Cas phosphorylation in a process that also requires the SH2 domain, leading to activation of signaling pathways that stimulate lamellipodial protrusion and ruffling. Step 3: Signaling is terminated by the CRL5$^{SOCS6}$-dependent ubiquitination and degradation of BCAR3 and Cas, phosphorylated in their respective phosphodegrons. Remaining questions include the identity of the kinase that phosphorylates BCAR3 Y117, the timing of Y117 phosphorylation, and the mechanism by which BCAR3 pY117 and SH2 domain stimulate Cas phosphorylation.

**Table 2.** Sequence alignment of SOCS6-binding sites.

| Protein | Residue | Sequence* | Evidence (reference) | | |
|---|---|---|---|---|---|
| | | | peptide† | mutant‡ | biology§ |
| Kit | pY568 | gnn**YVYI**dptq | (1,2) | (1) | |
| PDGFR | pY579 | ghe**YIYV**dpmq | (2) | | |
| Flt3 | pY591 | eyf**YV**d**F**reye | (3) | (3) | (3) |
| Flt3 | pY919 | eei**YI**i**M**qscw | (3) | (3) | (3) |
| Cas | pY664 | med**Y**d**YV**hlqg | | (4) | (4) |
| Dab1 | pY198 | dpv**Y**q**YI**vfea | | (5) | (5-6) |
| BCAR3 | pY117 | tve**YV**k**F**sker | | (7) | (7) |
| Spot array¶ | | xxx**YVYI**xxxx<br>xxx**YIFF**xxxx<br>xxx**YMWM**xxxx<br>xxx**YWVV**xxxx | (8) | | |

\* Local sequence aligned to phosphotyrosine (Y). Capital letters indicate residues at +1 to +3 positions that fit the consensus from the spot array.

† Phosphopeptide binds to SH2 domain in vitro.

‡ Decreased binding of Y to F mutant protein in cells.

§ Y to F mutant is stabilized or gains function in cells.

¶ A library of phosphopeptides randomized at positions +one to +three was selected by binding to the SOCS6 SH2 domain. Bound peptides were sequenced. Residues selected at each position are shown in decreasing order of preference. References: (1) (*Bayle et al., 2004*) (2) (*Zadjali et al., 2011*) (3) (*Kazi et al., 2012*) (4) (*Teckchandani et al., 2014*) (5) (*Feng and Cooper, 2009*) (6) (*Simó and Cooper, 2013*) (7) this work, (8) (*Krebs et al., 2002*).

Our results reveal that BCAR3 Y117 is required for both signaling and degradation. This means that BCAR3 activation triggers BCAR3 degradation in a negative feedback loop. The dual use of a single site for activation and degradation resembles Y198 of the neuron migration protein Dab1, which binds downstream signaling molecules and also recruits SOCS6/7 for CRL5-dependent degradation (*Simó and Cooper, 2013*). Mutating the SOCS-binding site in BCAR3 or Dab1 generates a non-functional protein that accumulates at higher level. In contrast, the SOCS6-binding site in Cas is not needed for downstream signaling, so mutating the site causes increased levels of an active protein and a dominant gain-of-function phenotype (*Teckchandani et al., 2014*).

Our finding that BCAR3 is subject to complex post-translational regulation may be helpful in reconciling the observation that BCAR3 over-expression in cell culture is oncogenic whereas increased BCAR3 RNA expression correlates with favorable outcomes in patients (*Guo et al., 2014*; *Wallez et al., 2012*; *Zhang et al., 2018*). Post-translational regulation of protein degradation means that BCAR3 protein levels may not correlate with RNA levels in patient samples. In addition to the phosphorylation-dependent mechanism investigated here, a previous study showed that TGFβ stimulates proteasomal degradation of BCAR3 (*Guo et al., 2014*), and over-expressed BCAR3 and Cas appear to stabilize each other in breast cancer cells, dependent on their mutual binding (*Wallez et al., 2014*). Changes in BCAR3 protein stability or phosphorylation state might influence BCAR3 activity in patients, but such regulation at the protein level might not be accounted for when correlating patient outcomes with gene expression. Thus, further study is required to understand mechanisms of BCAR3 regulation and how mis-regulation may contribute to disease progression.

## Materials and methods

### Plasmids

pMXpuroII and pMXpuroII-shCul5 plasmids were made as previously described (*Teckchandani et al., 2014*). pLKO.1-nontarget small hairpin RNA (shRNA) control vector (SHC002) and pLKO.1-shSOCS6 (TRCN0000284351) were purchased (Sigma Aldrich). pCAG-T7-mSOCS1, pCAG-T7-mSOCS2, pCAG-T7-mSOCS3, pCAG-T7-mSOCS4, pCAG-T7-mSOCS5, pCAG-T7-mSOCS6, pCAG-T7-mSOCS7, and pCAG-T7-mCisH were made as previously described (*Simó and*

*Cooper, 2013*; *Teckchandani et al., 2014*). All pCAG-T7-SOCS plasmids used in this work have LC-QQ mutations in the SOCS box to prevent binding to CRL5, and were made using PfuTurbo DNA polymerase to perform site-directed mutagenesis followed by DpnI digestion. pCAG-T7-mSOCS6 R407K was made as previously described (*Teckchandani and Cooper, 2016*). pCAG-T7-mSOCS6 ΔC was made by PCR amplifying codons 1–381 of mSOCS6 using a 5' primer complementary to the T7 tag and a 3' primer that inserts a stop codon and a NotI site after codon 381. The PCR product was inserted into pCAG-T7-SOCS6 by BamH1/NotI restriction digest and ligation.

pCMV-SPORT6-mBCAR3 was purchased from the Harvard Plasmid Database (MmCD00318547). pcDNA5-3xFlag-mBCAR3 FRT/TO was made by Gateway cloning. mBCAR3 was PCR amplified with flanking attB sites from pCMV-SPORT6-mBCAR3 and inserted into pDONR221 with a BP reaction (Thermo Fisher Scientific). pDEST-5'−3xFlag-pcDNA FRT/TO was a kind gift of Dr. Anne-Claude Gingras. pDONR221-mBCAR3 was moved into pDEST-5'−3xFlag-pcDNA FRT/TO with an LR reaction (Thermo Fisher Scientific). pLX304-hBCAR3-V5 was obtained from the human ORFeome version 8.1 Entry clone collection (*Yang et al., 2011*). pLX304-hBCAR3-F5-V5 was made by Gibson Assembly (New England Biolabs). A gBlock (Integrated DNA Technologies) containing BCAR3 Y42F, Y117F, Y212F, Y266F, and Y429F mutations was assembled into pLX304-hBCAR3-V5.

rtTA-N144 was a gift from Andrew Yoo (Addgene plasmid # 66810). pLTRE3G-SNAP-V5-hBCAR3 was made as follows. A gBlock containing sequence for SNAP-V5 tags was inserted into pLenti CMVTRE3G eGFP Blast (w818-1) (gift of Eric Campenau, Addgene #27568) at the AgeI restriction site using Gibson Assembly to make pLTRE3G-SNAP-V5-eGFP. hBCAR3 was PCR amplified with flanking AgeI and XbaI sites and inserted into AgeI and XbaI-digested pLTRE3G-SNAP-V5-eGFP to make pLTRE3G-SNAP-V5-hBCAR3. pLTRE3G-SNAP-V5-hBCAR3 mutants were made as follows. Single mutants Y42F, Y212F, Y266F, Y429F, R177K, and double mutant EE (L744E/R748E) were made by site-directed mutagenesis with Q5 polymerase (New England Biolabs) and Dpn1 digestion. The Y117F mutant was made by Gibson Assembly. A gBlock containing the BCAR3 Y117F mutation was assembled into pLTRE3G-SNAP-V5-hBCAR3 WT. pLTRE3G-SNAP-V5-hBCAR3 F5 was made by BamHI restriction digest of pLTRE3G-SNAP-V5-hBCAR3 Y42F, discarding the region between codon 107 and 763 and inserting the corresponding region from pLX304-hBCAR3-F5-V5. pLTRE3G-SNAP-V5-hBCAR3 F4 was made by PshAI restriction digest of pLTRE3G-SNAP-V5-hBCAR3 F5, discarding the region between codon 67 and 163 inserting the corresponding region from pLTRE3G-SNAP-V5-hBCAR3 WT.

## Cell lines

Parental cell lines were originally obtained from ATCC and confirmed by DNA fingerprinting using short tandem repeat, or STR, typing. Cell lines were screened periodically for mycoplasma using the MycoProbe kit from R and D Systems.

MCF10A cells were cultured in growth medium (DMEM/F12 (Thermo Fisher Scientific) supplemented with 5% horse serum (Thermo Fisher Scientific), 0.1 µg/ml cholera toxin (EMD Millipore), 10 µg/ml insulin (Thermo Fisher Scientific), 0.5 µg/ml hydrocortisone (Sigma-Aldrich), and 20 ng/ml EGF (Thermo Fisher Scientific)). Indicated experiments used assay media (DMEM/F12, 2% horse serum, 0.1 µg/ml cholera toxin, 10 µg/ml insulin, 0.5 µg/ml hydrocortisone, and 0 ng/ml EGF). The cells were passaged using 0.02% trypsin in PBS/EDTA. Trypsin was inactivated with an equal volume of DMEM/10% FBS and cells were harvested by centrifugation before resuspending in growth media. HeLa and 293 T cells were grown in DMEM/10% FBS and passaged with trypsin as above.

MCF10A empty vector (EV) and shCul5 cells were made as previously described (*Teckchandani et al., 2014*). MCF10A shControl and shSOCS6 cells were made as follows. Viruses containing pLKO.1-control and pLKO.1-shSOCS6 were packaged using HEK 293 T cells and MCF10A cells were infected. Stable cell lines were selected using 4 µg/mL puromycin.

*BCAR3*-/- MCF10A cells were made as follows. Early passage MCF10A cells were serially diluted and subclones were grown and characterized. Clone J8 was selected for its epithelial morphology. The J8 clonal cell line was infected with lentiCRISPR v2 (a gift from Feng Zhang; Addgene plasmid # 52961), lacking or containing guide RNA against BCAR3. pLCRISPRv2-sgRNA-hBCAR3(30) and (31) were made by cloning annealed oligos (5'-CACCGTCAGAGAGCTACCTGCCGAT-3' or 5'-CACCGCCCGAAACATACCAATCGGC-3') into the BsmBI site of plentiCRISPRv2. Potential knock-outs were isolated by single cell expansion. Validation of *BCAR3* knockout was done through genomic DNA isolation, PCR, and sequencing, as well as Western blotting.

MCF10A dox-inducible SNAP-V5 BCAR3 cells were made as follows. Viruses containing rtTA-N144 were packaged using 293 T cells and used to infect MCF10A cells. A stable line was selected using 50 µg/mL hygromycin. The MCF10A cell line stably expressing rtTA was then infected with viruses containing one of the pLTRE3G-SNAP-V5 constructs (pLTRE3G-SNAP-V5-eGFP, pLTRE3G-SNAP-V5-BCAR3 WT, Y42F, Y117F, Y212F, Y266F, Y429F, F5, F4, R177K, or EE). Stable lines were selected using 10 µg/mL blasticidin. Each cell line was hygromycin resistant and blasticidin resistant; however, not all cells within each line expressed the SNAP-V5-containing construct following induction with 50–100 ng/mL dox for 48–72 hr. To remove non-inducible cells, dox-treated cells were treated with 100 nM JaneliaFluor cp646JF-SNAP-ligand (Lavis Lab, Janelia Farms) (*Grimm et al., 2015*) for 1 hr, washed three times with PBS, incubated in ligand-free growth media for an hour and positive cells were sorted and harvested by FACS using the APC channel to detect cp646JF.

## Antibodies and reagents

| Antibody | Supplier | Catalog | Dilution |
|---|---|---|---|
| Rabbit anti-BCAR3 | Bethyl | A301-671A-M | Western (1:2000) |
| Rabbit anti-Cas (C-20) | Santa Cruz | sc-860 | Western (1:1000) |
| Mouse anti-Cas | BD Biosciences | 610271 | Western (1:5000) |
| Mouse anti-Cas | Santa Cruz | sc-20029 | IF (1:100) |
| Rabbit anti-pY165 Cas | Cell Signaling | 4015S | IF (1:200) |
| Rabbit anti-Cullin5 | Abcam | ab184177 | Western (1:1000) |
| Rabbit anti-pY397 FAK | Fisher | 44–624G | IF (1:200) |
| Rabbit anti-pY861 FAK | Fisher Scientific | 44–626G | IF (1:200) |
| Rabbit anti-GAPDH (0411) | Santa Cruz | sc-47724 | Western (1:1000) |
| Mouse anti-vinculin | Sigma | V9131 | Western (1:10000) |
| Mouse anti-paxillin | BD Biosciences | 610051 | IF (1:200) |
| Sheep anti-paxillin | R and D Systems | AF4259 | IF (1:200) |
| Mouse anti-FLAG | Sigma | F1804 | Western (1:1000) |
| Mouse anti-T7 | EMB Biosciences | 69522–4 | Western (1:5000) IP |
| Rabbit anti-V5 | Bethyl | A190-120A | Western (1:5000) IP |
| AlexaFluor 488 goat anti-rabbit IgG (H+L) | Invitrogen | A11008 | IF (1:1000) |
| AlexaFluor 647 goat anti-mouse IgG (H+L) | Invitrogen | A28181 | IF (1:1000) |
| AlexaFluor 647 goat anti-sheep IgG (H+L) | Invitrogen | A21448 | IF (1:1000) |

Reagents used: cycloheximide (Sigma), epidermal growth factor (Invitrogen), MLN4924 (Active Biochem), MG132 (Fisher Scientific), bafilomycin A, doxycycline (Fisher Scientific).

## Mass spectrometry

MCF10A EV and shCul5 cells were grown in 15 cm plates, two plates per condition, to approximately 80% confluence, washed in PBS, and lysed in 3 mL per plate of ice-cold 8 M urea containing 1 mM sodium orthovanadate. Proteins were reduced, alkylated and digested with trypsin as described (*Zhang et al., 2005*). Peptide labeling with iTRAQ reagents (TMT 6plex, Themo Fisher Scientific) was performed according to the manufacturer's instructions, using 400 µg of each sample. After reaction, the six samples were combined and concentrated under vacuum. For immunoprecipitation, protein G agarose (60 µl, Millipore) was incubated with anti-phosphotyrosine antibodies (12 µg 4G10 [Millipore], 12 µg PT66 [Sigma] and 12 µg PY100 [Cell Signaling Technology] for 8 hr at 4°C with rotation, then washed with 400 µl IP buffer [100 mM Tris HCl, 100 mM NaCl, and 1% Nonidet P-40, pH 7.4]). The TMT-labeled peptides were resuspended in 400 µl IP buffer and pH adjusted to 7.4, then mixed with the antibody-conjugated protein G agarose overnight at 4°C with rotation. The

antibody beads were spun down and the supernatant saved. The beads were washed with 400 μl IP buffer then four rinses with 100 mM Tris HCl pH 7.4 before elution with 70 μl of 100 mM glycine, pH 2.0 for 30 min at 25°C. Offline immobilized metal affinity chromatography (IMAC) was used to further enrich for phosphotyrosine peptides (*Zhang et al., 2005*).

Peptides were loaded on a precolumn and separated by reverse phase HPLC using an EASY-nLC1000 (Thermo) over a 140 min gradient from 100% solvent A ($H_2O/CH_3COOH$, 99:1 (v/v)) to 30% A: 70% B ($H_2O/CH_3CN/CH_3COOH$, 29:70:1 (v/v)). Eluted peptides were injected using nanoelectrospray into a QExactive Pluss mass spectrometer (Thermo) in data-dependent mode. The parameters for full scan MS were: resolution of 70,000 across 350–2000 *m/z*, AGC $3e^6$, and maximum IT 50 ms. The full MS scan was followed by MS/MS for the top 10 precursor ions in each cycle with a NCE of 34 and dynamic exclusion of 30 s. Raw mass spectral files (.raw) were searched using Proteome Discoverer (Thermo) and Mascot version 2.4.1 (Matrix Science). Mascot search parameters were: 10 ppm mass tolerance for precursor ions; 15 mmu for fragment ion mass tolerance; two missed cleavages of trypsin; fixed modifications were carbamidomethylation of cysteine and TMT 6plex modification of lysines and peptide N-termini; variable modifications were methionine oxidation, tyrosine, serine, and threonine phosphorylation. TMT quantification was obtained using Proteome Discoverer and isotopically corrected per manufacturer's directions, and normalized to the mean relative protein quantification ratios obtained from the total protein analysis (0.2% of the unbound peptides from the phosphotyrosine peptide immunoprecipitation). Mascot peptide identifications, phosphorylation site assignments and quantification were verified manually with the assistance of CAMV (*Curran et al., 2013*). Validated data were analyzed in Excel using a one-sided two-sample equal variance Student t-test. siRNA Transfection.

MCF10A cells were trypsinized and seeded in growth medium into a 12-well plate so as to be 50% confluent after attaching. A mixture of 50 pmol siRNA, 1.25 μl Lipofectamine 2000 (Invitrogen) in Optimem medium (Invitrogen) was added to the newly plated MCF10A cells and left for the cells to attach overnight. The next day the media was changed to fresh growth media. A second siRNA transfection was done 48 hr after the first transfection using the same protocol scaled up to use a six-well plate (125 pmol siRNA) or scaled down to use a 96-well plate (for migration assays). One day later, cells from the six-well dish were transferred to 4-well plates containing 12 mm coverslips for microscopy, or 12-well plates for protein analysis. Doxycline was added as needed at 10 ng/mL for 2–3 days before analysis. Cells were either lysed 48 hr after the second transfection or assays were performed as described.

| siRNA | Target sequence | Source | Notes | Catalog |
|---|---|---|---|---|
| Control siRNA | 5'-AATTCTCCCGAACGTGTCACGT-3' | Qiagen | | 1027310 |
| SOCS6 siRNA Pool | 5'-TAGAATCGTGAATTGACATAA-3'<br>5'-CAGCTGCGATATCAACGGTGA-3'<br>5'-TTGATCTAATTGAGCATTCAA-3'<br>5'-CGGGTACAAATTGGCATAACA-3' | Qiagen | | SI00061383<br>SI03068359<br>SI00061369<br>SI00061376 |
| Cul5 siRNA Pool | 5'-GACACGACGTCTTATATTA-3'<br>5'-CGTCTAATCTGTTAAAGAA-3'<br>5'-GATGATACGGCTTTGCTAA-3'<br>5'-GTTCAACTACGAATACTAA-3' | GE Dharmacon | | M-019553-01-0005 |
| BCAR1 siRNA Pool | 5'-AAGCAGTTTGAACGACTGGAA-3'<br>5'-CTGGATGGAGGACTATGACTA-3'<br>5'-CAGCATCACGCGGCAGGGCAA-3'<br>5'-CAACCTGACCACACTGACCAA-3' | Qiagen | | SI02757741<br>SI02757734<br>SI04438280<br>SI04438273 |
| BCAR3 siRNA Pool | 5'-CCGGAACTCTGGCGTCAACTA-3'<br>5'-CCGAGCGGCCACTCTGAGTAA-3'<br>5'-GCCCAACGAGTTTGAGTCAAA-3'<br>5'-AAGGTATCAGTTATATGATAT-3' | Qiagen | | SI03081603<br>SI03080196<br>SI00053102<br>SI00053095 |
| BCAR3 siRNA | 5'-GGUAACUACUGCUAAUGUUTT-3' | Life Technologies | Targets 3'UTR | AM16708 |

## qPCR

An RNeasy Plus Mini kit (Qiagen) was used to extract RNA and an iScript reverse transcription supermix (BioRad) was used to make cDNA. Control reactions lacked reverse transcriptase. cDNA

abundance was measured using an iTaq Universal SYBR Green Supermix kit (BioRad). Samples were run on the QuantStudio 5 Real-Time PCR System.

| Primer | Sequence |
| --- | --- |
| BCAR3 forward | 5' – AATCGCTTCTCCAAACAGAGC – 3' |
| BCAR3 reverse | 5' – ATTCACCGGCATGTTTCTGG – 3' |
| Cul5 forward | 5' – TTTTATGCGCCCGATTGTTTTG – 3' |
| Cul5 reverse | 5' – TTGCTGGGCCTTTATCATCCC – 3' |
| GAPDH forward | 5' – CAGCCTCAAGATCATCAGCA – 3' |
| GAPDH reverse | 5' – TGTGGTCATGAGTCCTTCCA – 3' |

## Cell lysis and immunoblotting

Cells were washed three times in phosphate-buffered saline (PBS) before lysis. Cells were lysed in X-100 buffer (1% Triton X-100, 150 mM NaCl, 10 mM HEPES pH 7.4, 2 mM EDTA, 50 mM NaF) or RIPA buffer (1% Triton X-100, 1% sodium deoxycholate, 0.1% SDS, 20 mM Tris-HCl pH 7.4, 150 mM NaCl, 5 mM EDTA, 5 mM EGTA) with fresh protease and phosphatase inhibitors (10 µg/mL Aprotinin, 1 mM PMSF, 1 mM sodium vanadate) added before use.

Lysates were adjusted to SDS sample buffer, heated to 95°C, resolved by SDS-PAGE using 10% polyacrylamide/0.133% bis-acrylamide gels, and transferred onto nitrocellulose membrane. The membrane was blocked in Odyssey blocking buffer (TBS) (LI-COR Biosciences) with 5% BSA for phosphotyrosine antibodies or 5% non-fat dry milk for all other antibodies. Following blocking, the membrane was probed with a primary antibody followed by IRDye 800CW goat anti-rabbit or 680RD goat anti-mouse conjugated secondary antibodies. Membranes were visualized using the Odyssey Infrared Imaging System (LI-COR Biosciences). Bands were quantified using ImageJ.

## DNA transfections and immunoprecipitation

HeLa cells were plated in six-well plates the day before transfection such that cells were 50% confluent on the day of transfection. A mixture of DNA, Lipofectamine 2000 (Thermo Fisher Scientific) and Optimem (Invitrogen) was made according to manufacturer's protocol, added to the cells and removed after 5 hr.

Immunoprecipitation experiments were conducted 24–48 hr after transfection. For indicated experiments, cells were incubated with 1 mM sodium pervanadate for 30 min. Cells were lysed on ice in X-100 buffer with fresh protease and phosphatase inhibitors (see above). Lysates were cleared by centrifugation for 10 min at 14,000 g. Lysates were rotated with 1 µg antibody at 4°C for 3 hr, after which Protein A/G plus agarose beads (Santa Cruz Biotechnology) were added for 1 hr at 4°C. Beads were collected by centrifugation and washed three times with X-100 buffer. The beads were resuspended in SDS sample buffer, boiled, lightly mixed to release bound protein and centrifuged. Immunoprecipitation samples were resolved by SDS-PAGE using 10% polyacrylamide gels and bound proteins detected by Western blotting as above. Samples of total cell lysate typically contained 5% of the protein used for immunoprecipitation.

## Cycloheximide chase

Cells were grown to 80% confluency and treated with 25 µg/mL cycloheximide by adding it directly to the conditioned media on the cells. Cells were treated with cycloheximide for 0, 2, 4, or 8 hr and lysed. Quantification of Western blots was done in ImageJ and BCAR3 protein levels were normalized to the loading control (GAPDH or vinculin).

## Scratch wound assay

The desired proteins were knocked down using two siRNA transfections, as previously described. Cells were plated in ImageLock 96-well dishes (Essen BioScience) for the second siRNA transfection and transfection materials were scaled down proportional to dish sizes. Cells were plated at 30% confluence at the time of the second transfection. For migration assays where doxycycline (dox)-induction was required, the media was replaced with growth media containing dox (10 ng/mL

except where noted) 6 hr after transfection and after two washes with PBS. For all migration assays, the confluent monolayers were placed in EGF-free assay media (with dox when required) 48 hr after the second transfection transfection. Monolayers were scratched using an Incucyte WoundMaker (Essen BioScience) 8 hr after being placed in assay media, the wells were washed with PBS three times to remove debris and cells were placed back in assay media (with dox when required). Scratch wounds were imaged once every 2 hr on an IncuCyte S3 and images were analyzed using the scratch wound function on the IncuCyte image analysis software. Overall migration was measured at 12 hr using the relative wound density calculated by the analysis software. Lamellipodium length was measured using the ruler in the IncuCyte image analysis software. Membrane ruffling was scored at 6 hr by counting the number of cells with ruffles relative to the total cell number. Ruffles were visualized as dark contrast at the front of the protrusion.

## Transwell migration and invasion assays

Cells were grown in EGF-free assay media for 24 hr before the assay. Migration assays were performed in 24-well chemotaxis chamber with an 8 µm pore size polyethylene terephthalate filter that separated the upper and lower wells (Thermo Fisher Scientific). Invasion assays were performed in 24-well Matrigel invasion chambers with an 8 µm pore size polyethylene terephthalate filter coated in Matrigel matrix that separated the upper and lower wells (Thermo Fisher Scientific). In both assays, the lower wells were filled with MCF10A growth media. A total of 80,000 cells were resuspended in EGF-free assay media and added to the top well. After 24 hr, cells on the top of the membrane were removed and the cells on the bottom of the membrane were fixed with methanol and stained with SybrSafe at 1:10,000 for 15 min. Membranes were rinsed, imaged and nuclei were counted.

## Immunofluorescence

For experiments that required protein knockdown, cells were transfected with siRNA as previously described. On the day following the second siRNA transfection, cells were plated on 12 mm #1.5 coverslips at 30% confluency in four-well (1.9 cm$^2$) plates (with dox when required). Confluent monolayers were transferred to assay media and incubated overnight before scratching with a P200 pipette tip. To detect the SNAP tag, JaneliaFluor cpJF549 SNAP-ligand (Lavis Lab, Janelia Farms) (*Grimm et al., 2015*) was added to the cells at 100 nM 4 hr after forming the wounds. Cells were incubated for 1 hr with the SNAP-ligand, washed three times with PBS and placed in fresh, ligand-free assay media for 1 hr to remove unreacted SNAP-ligand. Cells were fixed and permeabilized a total of 6 hr after the start of migration.

Fixation and immunostaining was done at room temperature. Cells were washed with PBS, fixed and permeabilized with 0.1% Triton X-100, 4% paraformaldehyde (PFA) in PBS for 2 min, and further fixed with 4% PFA/PBS for 15 min. Cells were blocked with 2% BSA, 5% normal goat serum in PBS for 30 min or overnight at 4°C. Primary antibodies were diluted in block solution (typically 1:200 dilution) and added for 3 hr. Coverslips were washed three times with PBS. AlexaFluor-labeled secondary antibodies, diluted 1:1000 in PBS, were added for 1 hr. Coverslips were washed three times with PBS, mounted in Prolong Glass Antifade Mountant and left to cure overnight in the dark before imaging or storing in the cold.

## Imaging and image quantification

Coverslips were imaged using 63x, 1.40 NA or 100x, 1.40 NA oil objectives on a Leica SP8 confocal microscope and deconvolved using Leica LAS acquisition software. In order to ensure unbiased data collection, 8–14 cells along the leading edge of the wound were imaged in the paxillin channel before imaging channels showing other antigen(s) of interest. The same scale and image settings were used for all conditions within an experiment. The number of cells and biological replicates is provided in each figure legend.

Image quantification was done in ImageJ. A line was drawn ~6 µm from the front of the migrating cells, as described (*Teckchandani and Cooper, 2016*), to isolate the leading edge. Using the threshold tool on ImageJ, paxillin-containing adhesions were identified and mean intensity of SNAP or other antigens (pY-Cas, Cas, pY397-FAK, pY-861-FAK and paxillin) was measured. The relative abundance of each antigen in adhesions was calculated by dividing the mean antigen intensity by the

mean intensity of paxillin. Because expression of SNAP-V5-BCAR3 varied slightly cell by cell, we corrected for expression level as follows. SNAP intensity was measured in the leading edge adhesions ($SNAP_{ad}$) and rear ($SNAP_{rear}$) of dox-induced cells and in non-induced cells ($SNAP_{bg}$). Corrected SNAP intensity in adhesions ($SNAP_{ad}$-$SNAP_{bg}$)/($SNAP_{rear}$-$SNAP_{bg}$) was divided by the mean paxillin intensity.

## Acknowledgements

We are most grateful to Forest White for advice on quantitative phosphotyrosine proteomics and Amanda Del Rosario of the Koch Institute Proteomics Core for running the analysis. Julio Vazquez, Dave McDonald, Lena Schroeder and Peng Guo of the FHCRC Cellular Imaging Shared Resource provided outstanding assistance with microscopy. We thank Patrick Paddison, Anne-Claude Gingras and Susumu Antoku for plasmids and Luke Lavis for Janelia Fluors. Laura Arguedas-Jiminez provided excellent technical assistance. We are grateful to Saurav Kumar and Dayoung Kim for discussion and helpful comments on the paper. ES was supported in part by Predoctoral Institutional Training Program T32 GM007270 to the University of Washington. This work was supported by research grant R01 GM109463 from the US Public Health Service and by discretionary funds from the Fred Hutchinson Cancer Research Center. The FHCRC Flow Cytometry and Cellular Imaging Shared Resources are supported by P30 CA015704. The Koch Institute Proteomics Core is supported by P30 CA14051.

## Additional information

### Competing interests

Jonathan A Cooper: Senior editor, *eLife*. The other authors declare that no competing interests exist.

### Funding

| Funder | Grant reference number | Author |
| --- | --- | --- |
| National Institutes of Health | T32 GM007270 | Elizabeth M Steenkiste |
| National Institutes of Health | R01 GM109463 | Elizabeth M Steenkiste<br>Jason D Berndt<br>Carissa Pilling<br>Christopher Simpkins<br>Jonathan A Cooper |
| National Institutes of Health | P30 CA015704 | Elizabeth M Steenkiste<br>Jason D Berndt<br>Carissa Pilling<br>Christopher Simpkins<br>Jonathan A Cooper |

The funders had no role in study design, data collection and interpretation, or the decision to submit the work for publication.

### Author contributions

Elizabeth M Steenkiste, Conceptualization, Investigation, Writing - original draft, Writing - review and editing; Jason D Berndt, Conceptualization, Investigation, Writing - review and editing; Carissa Pilling, Conceptualization, Data curation, Investigation; Christopher Simpkins, Investigation, Writing - review and editing; Jonathan A Cooper, Conceptualization, Data curation, Supervision, Funding acquisition, Investigation, Writing - original draft, Writing - review and editing

### Author ORCIDs

Elizabeth M Steenkiste http://orcid.org/0000-0001-6452-2340
Christopher Simpkins http://orcid.org/0000-0003-3174-6609
Jonathan A Cooper https://orcid.org/0000-0002-8626-7827

Decision letter and Author response
Decision letter https://doi.org/10.7554/eLife.67078.sa1
Author response https://doi.org/10.7554/eLife.67078.sa2

## Additional files

### Supplementary files

• Source data 1. Source data for cropped images of gel blots. Except where noted, blots were probed with anti-rabbit 800 and anti-mouse 700 and scanned on a Odyssey Infrared Imaging System. Individual files include lane designation and a brief explanation of antibodies used. Rb, rabbit. Ms, mouse.

• Supplementary file 1. Quantification of phosphotyrosine peptides in control and Cul5-deficient MCF10A cells. Phosphopeptides whose abundance significantly differed between control and Cul5-deficient MCF10A cells in two separate experiments. Sheet 1: Experimental conditions, numbers of phosphopeptides quantified, and number of significant changes. Sheet 2: Phosphopeptides that increased significantly in Experiment 1. Sheet 3: Phosphopeptides that increased significantly in Experiment 2. Sheet 4: Phosphopeptides that increased significantly in both Experiments (see also *Table 1*).Source data (.raw files and Excel files with ratios and statistics) are available at https://doi.org/10.5061/dryad.tmpg4f4zn

• Transparent reporting form

### Data availability

All data generated or analysed during this study are included in the manuscript and supporting files, with the exception of the raw mass spectrometry data, which have been deposited in the Dryad Digital Repository.

The following dataset was generated:

| Author(s) | Year | Dataset title | Dataset URL | Database and Identifier |
|---|---|---|---|---|
| Cooper JA | 2021 | Data from: Phosphotyrosine peptide abundance in control and Cul5-deficient MCF10A cells | https://doi.org/10.5061/dryad.tmpg4f4zn | Dryad Digital Repository, 10.5061/dryad.tmpg4f4zn |

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
