## [Decision Letter]

**Acceptance summary:**

This work will be of particular interest to cell and cancer biologists interested in the molecular regulation of cell migration. It sheds new light on the regulation and function of a signaling network comprised of the adaptor molecules Cas and BCAR3 that functions to control cell motility. The quite complex data presented in the manuscript are generated through rigorous experimentation, frequently with the use of multiple approaches to arrive at the strong stated conclusions.

**Decision letter after peer review:**

Thank you for submitting your article "A Cas-BCAR3 co-regulatory circuit controls lamellipodia dynamics" for consideration by *eLife*. Your article has been reviewed by 3 peer reviewers, and the evaluation has been overseen by a Reviewing Editor and Anna Akhmanova as the Senior Editor. The following individual involved in review of your submission has agreed to reveal their identity: Michael Sheetz (Reviewer #2).

Essential revisions:

Summary:

This study focuses on the formation of adaptor protein complexes at adhesion sites and their links to in vitro membrane ruffling and cell movement. Specifically, the authors study the role of the adaptor BCAR3 protein which is regulated by post-translational mechanisms (ubiquitin degradation and tyrosine phosphorylation). Experiments are generally performed using transiently-transfected MCF10A or Hela cells to knock down (mainly via siRNA) or over-express tagged proteins. Using proteomics, a new phosphorylation site was identified (BCAR3 Y117) under conditions of pervanadate stimulation or in EGF-containing MCF10A growth media. Mutagenesis showed that BCAR3 Y117 is potentially important for enhancement of in vitro cell movement under conditions where the cullin-5 E3 ligase has also been reduced by siRNA expression. The authors propose a "co-regulatory" model whereby the recruitment of BCAR3 to adhesions acts to modulate p130Cas tyrosine phosphorylation and cell migration.

If the case is strengthened as suggested below, would extend our understanding of molecular regulation of cell migration and should be of interest to cell and cancer biologists. The data are consistent with the conclusions, but overall the case for cause and effect is not yet considered compelling by all reviewers, and the cell motility experiments are not causally connected to a specific signaling event.

There are some unresolved issues where new data, or further explanations, will improve support for the central claims of the paper.

1. The cell phenotypes are only been shown when cullin-5 (CUL5) is knocked down or BCAR3 (mutants) are overexpressed, both of which obviously result in non-physiological levels of expression. CUL5 loss will almost certainly alter multiple aspects of cell signalling (see point 5 below).

2. If it is true that a "single phosphorylation site in BCAR3 for activation and degradation", then would the expression of this mutant be expected to have a dominant-negative or activating phenotype? Experiments overexpressing BCAR3 mutations at this site are lacking comparisons with other BCAR3 Y to F mutants that retain "activity" – all BCAR3 mutants shown are loss of function. What about comparison with the 4F mutant, for example?

3. As multiple protein domains and post-translational modifications are involved in the BCAR3-p130Cas complex, the authors have not completely established a convincing mechanistic linkage between the newly-identified BCAR3 Y117 phosphorylation, SOCS6 binding, and a CUL5-dependent cell phenotype.

4. Some of the experimental conditions (+/- EGF in growth media) are not optimal to make connections with EGFR activation and signalling downstream.

5. Is tyrosine phosphorylation of Y117 an EGF-stimulated event or just detected after phosphatase inhibition with pervanadate? The activation of Abl kinase (ABL1 increase at Y393, an activation site) in CUL5 deficient cells is possibly indicative of multiple signaling pathway activation and not necessary a Src effect.

6. The introduction / use of the BCAR3 L744E/R748E mutant needs further discussion/explanation. Could this mutation not have effects other than disrupting p130Cas binding? Figure 3 and connections to SOCS6 are confusing; the authors state that "SOCS6 binds BCAR3 and Cas independently" (bottom of page 7). However, while they show that the EE BCAR3 mutant binds to SOCS6 under conditions when it does not bind to Cas, they do not show the reciprocal interaction in this paper. In the data shown in Figure 5B, BCAR3-EE mutant expression is relatively low and this makes the loss of binding conclusion weak.

7. Regarding Figure 7, can the authors comment on potential reasons for why their data with regards to the inability of the EE mutant of BCAR3 to localize to focal adhesions may differ from other published data showing that a similar BCAR3 mutant was seen to efficiently localize to adhesions where it exhibited differences in the dynamics of BCAR3 dissociation from the adhesions (Cross et al., Oncogene. 2016)?

8. The use of the pY165-Cas antibody to measure changes in p130Cas activation at adhesions (Figure 8) needs further controls as the signal is relatively weak. Missing is the total p130Cas levels at paxillin-containing adhesions. How do the authors interpret the negative effects of the BCAR3 R177K or BCAR3-EE mutants on p130Cas phosphorylation? These experiments are correlative and not a strong mechanistic cause-effect ending for the paper.

9. Is it known whether it is the phosphorylation at Y165 within the pool of Cas that is localized to adhesions that is reduced in the presence of Y117F, R177K, or the EE mutants of BCAR3?

---

## [Author Response]

Summary:This study focuses on the formation of adaptor protein complexes at adhesion sites and their links to in vitro membrane ruffling and cell movement. Specifically, the authors study the role of the adaptor BCAR3 protein which is regulated by post-translational mechanisms (ubiquitin degradation and tyrosine phosphorylation). Experiments are generally performed using transiently-transfected MCF10A or Hela cells to knock down (mainly via siRNA) or over-express tagged proteins.

We did indeed use transient transfection to over-express tagged proteins in HeLa cells for the binding experiments in Figures 3 and 4, but were at pains to carefully titrate expression levels using doxycyline-inducible lentiviral MCF10A cell lines in the functional assays. We will address expression levels further in response to point 1 below.

Using proteomics, a new phosphorylation site was identified (BCAR3 Y117) under conditions of pervanadate stimulation or in EGF-containing MCF10A growth media.

The proteomics was done on cells cultured either in EGF-containing growth media or the standard MCF10A assay media, which lacks EGF. No pervanadate was used. We will address the role of EGF further in response to points 4 and 5 below.

Mutagenesis showed that BCAR3 Y117 is potentially important for enhancement of in vitro cell movement under conditions where the cullin-5 E3 ligase has also been reduced by siRNA expression. The authors propose a "co-regulatory" model whereby the recruitment of BCAR3 to adhesions acts to modulate p130Cas tyrosine phosphorylation and cell migration.If the case is strengthened as suggested below, would extend our understanding of molecular regulation of cell migration and should be of interest to cell and cancer biologists. The data are consistent with the conclusions, but overall the case for cause and effect is not yet considered compelling by all reviewers, and the cell motility experiments are not causally connected to a specific signaling event.There are some unresolved issues where new data, or further explanations, will improve support for the central claims of the paper.

We are pleased that the reviewers agree that our study has the potential to extend our understanding of regulation of cell migration, but disappointed that the cause and effect were not brought out well. In revising the paper, we have taken extra care with the Abstract, Introduction and Discussion to better explain the previous publications and highlight the new insights afforded by our research.

1. The cell phenotypes are only been shown when cullin-5 (CUL5) is knocked down or BCAR3 (mutants) are overexpressed, both of which obviously result in non-physiological levels of expression. CUL5 loss will almost certainly alter multiple aspects of cell signalling (see point 5 below).

In the initial submission, we reported that BCAR3 is needed for collective migration only when Cul5 is absent (Figure 2d). Our subsequent experiments used Cul5-deficient cells to assay rescue of collective migration by BCAR3 mutants (Figure 6c). In these conditions, the ectopic BCAR3 was over-expressed 2- to 3-fold. We agree this may be non-physiological. However, we also showed in Figure 2b that single cell migration requires BCAR3 whether or not Cul5 is present. This provides us with an assay for BCAR3 function that does not require Cul5 knockdown. In the revised manuscript, we have added results showing rescue of single-cell migration by BCAR3 mutants in Cul5-containing cells (Figure 6b). The results confirm that the domains and residues of BCAR3 required for collective migration in the absence of Cul5 (Figure 6c) are also required for single-cell migration when Cul5 is present (Figure 6b). We thank the reviewers for bringing this to our attention. Adding the single cell migration data strengthens the case that BCAR3 functions require Tyr 117, the SH2 domain, and Cas binding. In addition, please note that all the immunofluorescence in the old paper (and new immunofluorescence experiments in the revised paper) were done without Cul5 knockdown or BCAR3 over-expression. Therefore we are confident that the mechanisms described here are physiologically relevant.

2. If it is true that a "single phosphorylation site in BCAR3 for activation and degradation", then would the expression of this mutant be expected to have a dominant-negative or activating phenotype? Experiments overexpressing BCAR3 mutations at this site are lacking comparisons with other BCAR3 Y to F mutants that retain "activity" – all BCAR3 mutants shown are loss of function. What about comparison with the 4F mutant, for example?

Thank you for the suggestion. We have now performed experiments with the 4F mutant and added the results to Figure 5c, Figure 5 —figure supplement 1, Figure 6 (all panels), Figure 8, and Figure 8 —figure supplement 3. In all cases F4 behaves like wildtype. This shows that the other four tyrosine phosphorylation sites play little if any role in regulating BCAR3 degradation, localization or function.

3. As multiple protein domains and post-translational modifications are involved in the BCAR3-p130Cas complex, the authors have not completely established a convincing mechanistic linkage between the newly-identified BCAR3 Y117 phosphorylation, SOCS6 binding, and a CUL5-dependent cell phenotype.

We hope the revised text and new experiments make the molecular mechanism clearer, but agree that we don’t completely understand the mechanism at this time. The challenge is that Y117 has positive and negative effects, activating Cas and cell motility on the one hand but also inducing BCAR3 turnover on the other. In addition, Cul5 has other substrates. Indeed, Cul5 and SOCS6 target both Cas and BCAR3. This makes it challenging to design experiments that would isolate effects of Cul5 on BCAR3 from effects on other substrates. A full understanding of molecular mechanism may require a full structural analysis.

4. Some of the experimental conditions (+/- EGF in growth media) are not optimal to make connections with EGFR activation and signalling downstream.

We do not know whether EGFR regulates BCAR3. We initially detected BCAR3 Y117 phosphorylation by phosphoproteomics of MCF10A cells grown in either EGF-containing growth media or EGF-deficient assay media (Figure 1 —figure supplement 1). In addition, Cul5 regulated BCAR3 protein but not RNA levels in both assay and growth media (Figure 1c). Since Cul5-mediated turnover of BCAR3 requires pY117, this is consistent with phosphorylation of Y117 whether EGF is present or absent. However, we have not identified the kinase and cannot rule out the EGFR (discussed in paragraph starting at foot of page 11).

5. Is tyrosine phosphorylation of Y117 an EGF-stimulated event or just detected after phosphatase inhibition with pervanadate?

As discussed above (point 4), EGF appears not to regulate Y117 phosphorylation. Phosphatase inhibition with pervanadate was not needed to detect pY117 in the phosphoproteomics experiments (Figure 1 —figure supplement 1) or to detect coprecipitation of SOCS6 and BCAR3 (Figure 4a). However, pervanadate was used to increase SOCS6-BCAR3 co-immunoprecipitation in Figure 3 and 4b-d.

The activation of Abl kinase (ABL1 increase at Y393, an activation site) in CUL5 deficient cells is possibly indicative of multiple signaling pathway activation and not necessary a Src effect.

We were also intrigued by the increase in Abl pY393, as well as phosphosites in the activation loop of the insulin/IGF1 receptor, in Cul5-deficient cells. These sites are all known Src-family kinase sites as well as autophosphorylation sites. Follow up experiments revealed that the steady state levels of Abl and IGF1R did not change when Cul5 is removed. Inhibiting Abl with STI571/Gleevec did not inhibit the increased migration of Cul5-deficient cells, implying that BCAR3 Y117 was still phosphorylated. However, at this stage, we are not confident ruling in or out some kind of kinase relay.

6. The introduction / use of the BCAR3 L744E/R748E mutant needs further discussion/explanation. Could this mutation not have effects other than disrupting p130Cas binding?

We agree that the EE mutant could have other effects. Other investigators have found that this mutation inhibits binding to Cas and has the same effect on cell proliferation as mutating the interacting residues in Cas (Wallez et al., 2014). The fact that the EE mutation and Cas knockdown both prevent BCAR3 localization to adhesions is most consistent with BCAR3 localizing to adhesions by binding to Cas.

Figure 3 and connections to SOCS6 are confusing; the authors state that "SOCS6 binds BCAR3 and Cas independently" (bottom of page 7). However, while they show that the EE BCAR3 mutant binds to SOCS6 under conditions when it does not bind to Cas, they do not show the reciprocal interaction in this paper.

We agree that we did not show that Cas binds SOCS6 in the absence of BCAR3. This was an extrapolation from our previous identification of a SOCS6-binding site in Cas that is distinct from the BCAR3-binding site (Teckchandani et al., 2014). However, in the absence of a direct test, we have changed the text in the Results and Discussion, omitting “independently” when referring to Cas degradation. We have also changed the title of the Results section on page 7 from: “CRL5 separately targets BCAR3 and Cas through SOCS6” to “CRL5 directly targets BCAR3 through SOCS6”.

In the data shown in Figure 5B, BCAR3-EE mutant expression is relatively low and this makes the loss of binding conclusion weak.

We agree Figure 5b was not convincing due to low expression of BCAR3^EE^. We have therefore repeated the co-immunoprecipitations and the lack of binding to Cas is more obvious. The new result has been moved to Figure 5 —figure supplement 1.

7. Regarding Figure 7, can the authors comment on potential reasons for why their data with regards to the inability of the EE mutant of BCAR3 to localize to focal adhesions may differ from other published data showing that a similar BCAR3 mutant was seen to efficiently localize to adhesions where it exhibited differences in the dynamics of BCAR3 dissociation from the adhesions (Cross et al., Oncogene. 2016)?

We would prefer not to speculate. The absence of BCAR3^EE^ from adhesions agrees with the absence of BCAR3 from adhesions when Cas is knocked down. Perhaps the difference is due to cell type. (Cross et al., 2016) found that GFP-tagged BCAR3^EE^ co-localized with Cas in adhesions in BT549 cells. However, (Wallez et al., 2014) found that BCAR3^EE^ did not co-localize with Cas in membrane ruffles on MCF7 cells. Prior papers used single mutants that did not effectively inhibit BCAR3-Cas binding. We all agree that BCAR3 needs to bind Cas for biological activity. The BT549 cells are aggressive breast cancer cells and MCF10A cells are normal. Perhaps other proteins recruit BCAR3^EE^ to adhesions in BT549 cells.

8. The use of the pY165-Cas antibody to measure changes in p130Cas activation at adhesions (Figure 8) needs further controls as the signal is relatively weak.

We agree that we need extra controls to show pY165-Cas antibody specificity for immunofluorescence. New Figure 8 —figure supplement 1 shows that pY165-Cas fluorescence signal is inhibited when Cas is knocked down or when Src family kinases are inhibited. We get similar results with a pY410-Cas antibody, which recognizes another pYxxP motif in Cas, but did not include the data.

Missing is the total p130Cas levels at paxillin-containing adhesions.

When we initially submitted the paper, we had tested two different total Cas antibodies and neither was useful for immunofluorescence. We have now purchased a different anti-Cas antibody and get good signals for Cas in paxillin-containing adhesions. Using this antibody, we have confirmed our conclusion that BCAR3 is not needed for Cas localization to adhesions. The previous experiment made use of EYFP-Cas (previous Figure 7a). The new Figure 7a shows that endogenous Cas localizes to adhesions in the absence of BCAR3. We also added Figure 8 —figure supplement 3, showing that endogenous Cas localization to adhesions is not altered when endogenous BCAR3 is removed and BCAR3 wildtype or mutants are re-expressed. Unfortunately, both the pY165-Cas and total Cas antibodies are rabbit so we cannot stain for both at the same time, but the pY165-Cas/paxillin and Cas/paxillin ratios make it clear that the loss-of-function BCAR3 mutants affect Cas phosphorylation, not localization.

How do the authors interpret the negative effects of the BCAR3 R177K or BCAR3-EE mutants on p130Cas phosphorylation? These experiments are correlative and not a strong mechanistic cause-effect ending for the paper.

The lack of rescue by BCAR3^EE^ is easy to explain since it does not bind Cas and is absent from adhesions, so has no effect. It’s true that we don’t know why BCAR3^R177K^ is inactive. We assume it interferes with SH2 domain function but do not know the ligand. Obvious candidates are the EGFR and PTPRA. (Oh et al., 2008) reported that BCAR3 co-immunoprecipitates with EGFR and (Jones et al., 2006) reported that the BCAR3 SH2 domain binds a ErbB2 pY1139 phosphopeptide with K_d_ ~ 1 uM. Sun et al. showed co-precipitation of full length BCAR3 and RPTPA from transfected cells that was abolished by mutation of the BCAR3 SH2 domain or mutation of an pY site in PTPRA. However, we probed SNAP-V5-BCAR3 immunoprecipitates with antiphosphotyrosine, anti-EGFR and anti-PTPRA antibodies to look for possible co-precipitating bands and saw none. We’ve also tried making the BCAR3 SH2 domain according to (Oh et al., 2008) and it was insoluble.

To avoid ending the paper on a negative note, we have added a new result. We found that BCAR3 is needed not only to phosphorylate Cas Y165 but also to phosphorylate FAK Y861, another SFK phosphorylation site in integrin adhesion complexes. FAK autophosphorylation at Y397 was unaffected (Figure 8 —figure supplement 2). The implication is that BCAR3 regulates SFKs in adhesions.

9. Is it known whether it is the phosphorylation at Y165 within the pool of Cas that is localized to adhesions that is reduced in the presence of Y117F, R177K, or the EE mutants of BCAR3?

In the absence of acute stimulation, the fraction of Cas that is phosphorylated at Y165 is very small and not affected by BCAR3 depletion. We don’t know the status of Cas phosphorylation in the cytoplasm, but our data in Figure 8 shows that phosphorylation stoichiometry of the important population of Cas in adhesions is regulated by BCAR3 dependent on Y117, R177 and the Cas-binding site on BCAR3. We think that the focus on Cas in adhesions is more relevant to the biology.